# EgoPrivacy: What Your First-Person Camera Says About You?

**Yijiang Li** [1]  **Genpei Zhang** [2]  **Jiacheng Cheng** [1]  **Yi Li** [3]  **Xiaojun Shan** [1]  **Dashan Gao** [3]  **Jiancheng Lyu** [3]
**Yuan Li** [3]  **Ning Bi** [3]  **Nuno Vasconcelos** [1]

## Abstract

While the rapid proliferation of wearable cameras has raised significant concerns about egocentric video privacy, prior work has largely overlooked the unique privacy threats posed to the camera wearer. This work investigates the core question: *How much privacy information about the camera wearer can be inferred from their first-person view videos?* We introduce EgoPrivacy, the first large-scale benchmark for comprehensive evaluation of privacy risks in egocentric vision. EgoPrivacy covers three types of privacy (demographic, individual, and situational) defining seven tasks that aim to recover private information ranging from fine-grained (e.g., wearer's identity) to coarse-grained (e.g., age group). To further emphasize the privacy threats inherent to egocentric vision, we propose *Retrieval-Augmented Attack*, a novel attack strategy that leverages ego-to-exo retrieval from an external pool of exocentric videos to boost the effectiveness of demographic privacy attacks. An extensive comparison of the different attacks possible under all threat models is presented, showing that private information of the wearer is highly susceptible to leakage. For instance, our findings indicate that foundation models can effectively compromise wearer privacy even in zero-shot settings by recovering attributes such as identity, scene, gender, and race with 70–80% accuracy. Our code and data are available at https://github.com/williamium3000/ego-privacy.

## 1. Introduction

The growing adoption of wearable cameras and egocentric (first-person view) videos, driven by advances in hardware and computer vision (Betancourt et al., 2015; Plizzari et al., 2024; Sigurdsson et al., 2018a; Grauman et al., 2022; 2024), enables innovative applications like activity recognition (Nguyen et al., 2016), human behavior analysis (Cazzato et al., 2020), or life logging (Bolanos et al., 2016; Del Molino et al., 2016). However, it also raises significant privacy concerns (Hoyle et al., 2014; 2015b). An already popular concern is the privacy of people *captured by* egocentric cameras (Farringdon & Oni, 2000; Krishna et al., 2005; Mandal et al., 2014; Chakraborty et al., 2016; Templeman et al., 2014; Korayem et al., 2016; Dimiccoli et al., 2018; Hasan et al., 2017; Fergnani et al., 2016). This concern, however, is not specific to egocentric video. Third-person cameras are already common in public environments, e.g. surveillance networks, and many private environments, e.g. TV sets with user facing cameras, motivating a line of research on privacy preserving cameras (Hinojosa et al., 2021; 2022; Cheng et al., 2024a; Khan et al., 2024) and post-hoc privacy techniques, e.g. methods to delete or obfuscate faces in images (Criminisi et al., 2003; 2004; Bitouk et al., 2008; Ren et al., 2018). While sharing all these issues, egocentric video introduces a new set of privacy concerns of its own, namely the privacy implications for the camera *wearers*, which have been much less studied (Hoshen & Peleg, 2016; Thapar et al., 2020a;b; Tsutsui et al., 2021).

*Wearer-centric* privacy is particularly concerning because egocentric videos are highly personal, captured continuously to document the day-to-day experience and surroundings of the camera wearer, and to keep track of their activities (Plizzari et al., 2024). The availability of this information will create pressures for its sharing, e.g. free video storage in exchange for video mining access, analysis by third parties, e.g. insurance companies collecting health information, and cross-referencing of egovideo with publicly available third-person video of the wearer, e.g. on social media platforms. All privacy problems currently posed by location-tracking apps will be magnified by the ability to know not only where people are but also *what they are doing* (Hoyle et al., 2014; Price et al., 2017; Speciale et al., 2019). All of this can lurk under a false sense of privacy, due to the fact that the

---
[1]University of California, San Diego [2]Carnegie Mellon University [3]Qualcomm AI Research, an initiative of Qualcomm Technologies, Inc. Correspondence to: Jiacheng Cheng <jiacheng.cheng96@gmail.com>.

*Proceedings of the 42$^{nd}$ International Conference on Machine Learning*, Vancouver, Canada. PMLR 267, 2025. Copyright 2025 by the author(s).

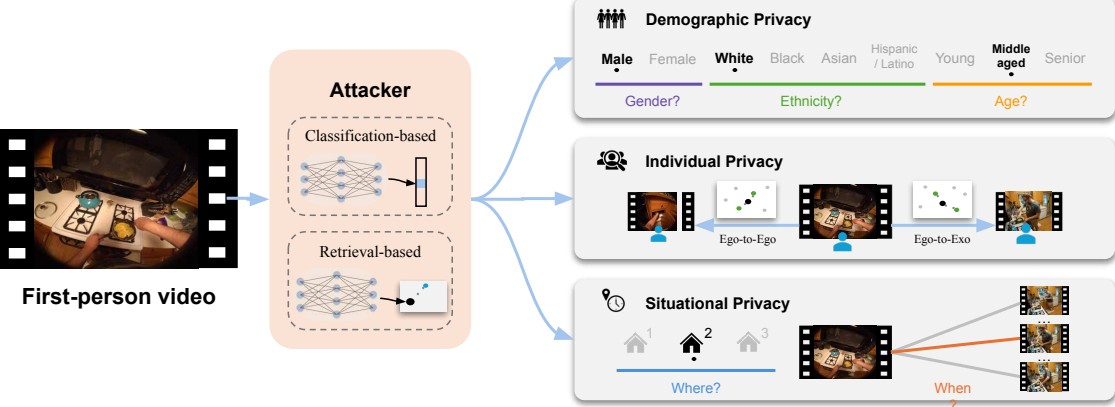

*Figure 1.* **Overview of the proposed EgoPrivacy benchmark.** What can you tell about the camera wearer from egocentric videos alone? It may come as a surprise that a fair amount of information about the user, such as demographics, identity, time and location of recording, can be inferred from their first-person view footages, despite not revealing their faces or full body.

camera is not framing its user. Given the limited attention to the problem, it is currently not even well understood *how much* of a privacy problem egocentric video poses to camera wearers. Questions such as what type of private information and how much of it can be recovered remain largely unanswered.

This work is a first attempt to define the range of *wearer-centric* privacy problems arising from egocentric recordings. In essence, we ask: *What can be told about the camera wearer by watching egocentric videos?* Figure 1 illustrates a variety of personal information that can be inferred from the video: hand appearance and pose can give away the gender, race and age of the wearer; egocentric videos can be matched to exocentric views of the wearer to fully reveal identity or activities; background settings and objects can give away location and activity; video clips can be matched to reason about location and time, and so forth. We group these privacy issues into three broad categories: *demographic* privacy for recognizing demographic groups of the wearer, *individual* privacy for uniquely identifying the wearer, and *situational* privacy for recognizing when and where the recording took place.

To comprehensively study the problem of egovideo privacy, we propose a novel large-scale benchmark, **EgoPrivacy**, annotated to allow the quantification of privacy risks under each of these categories. EgoPrivacy covers seven tasks representative of the three privacy categories, each formulated as either a problem of video classification or retrieval. We then propose a set of threat models with increasing levels of access to wearer data and perform an extensive evaluation of their ability to recover private information, using various types of foundation models.

Extensive experiments reveal significant privacy challenges, as *all* threat models are able to extract surprisingly high amounts of private information. For example, zero-shot foundation models are shown to have a remarkable ability

to compromise demographic privacy. This implies that even an adversary with no additional data or information about the wearer, can simply use open source models to recover attributes like race and gender. Fine-tuning these models on annotated exocentric or egocentric datasets extends this ability to recover attributes like wearer identity or scene location.

The gap between privacy attacks on egocentric and exocentric video largely owes to a key advantage of egocentric footage: it naturally hides the wearer's face and most parts of the body which can easily give away the privacy information of a subject. However, in practice, as almost everyone is increasingly exposed to all kinds of cameras in public, it is entirely possible that the camera wearer of an exocentric video will also be filmed in exocentric videos by a third part (e.g. suveilance systems, vloggers) simultaneously. If an adversary could get access to a repository of third-person view videos and successfully recover those third-person view corresponding to the ego video query, the risk of privacy leakage in egocentric vision will be elevated another level. Motivated by this, we introduce the novel *Retrieval-Augmented Attack* (RAA): With access to a repository of third-person videos that may feature the target user, an attacker first conducts ego-to-exo retrieval, then launches the privacy attack from the exocentric perspective. Experiments show that merging cues from the egocentric stream with the retrieved exocentric clip markedly raises the success rate of demographic-privacy attacks.

The gap between privacy attacks on egocentric and exocentric video can be attributed to a key advantage of egocentric footage: it naturally obscures the wearer's face and much of their body, that typically reveal private information. However, in practice, individuals are increasingly exposed to various public-facing cameras, making it highly plausible that the wearer of an egocentric camera is simultaneously captured in third-person view footages, e.g. by surveillance

systems or bystanders recording with personal devices. This scenario is far from hypothetical. For instance, consider a case where someone uploads a series of egocentric videos to social media. An attacker could potentially obtain the poster's IP address and retrieve surveillance footage from nearby locations. Motivated by this, we propose a novel Retrieval-Augmented Attack (RAA): the adversary first performs ego-to-exo retrieval to identify third-person clips containing the target, then launches a privacy attack from the exocentric perspective. Our experiments demonstrate that incorporating cues from retrieved third-person views into the analysis of egocentric footage significantly improves the effectiveness of demographic privacy attacks.

Overall, this paper makes four key contributions. First, we develop the first comprehensive large-scale benchmark for studying privacy in egocentric videos, which covers risks at the demographic, individual, and situational levels. Second, we formulate various threat models based on attacks with varying levels of access to video of the wearer and instantiate concrete attacker models for each of them. Third, we present an empirical analysis of the success of these attacks, revealing that even the use of zero-shot foundation models can suffice to expose significant amounts of private information. Last but not least, we further derive a novel privacy attack by ego-to-exo retrieval augmentation and demonstrate its effectiveness at exposing demographic attributes. We hope that our work can lay the foundation for future investigations into both offensive and defensive strategies concerning egocentric privacy.

## 2. Related Works

**Visual Privacy Benchmarks.** Large-scale public benchmarks are indispensable for successful computer vision research. Multiple benchmarks with privacy annotations (e.g. PIPA (Zhang et al., 2015), VISPR (Orekondy et al., 2017), VizWiz-Priv (Gurari et al., 2019)) have been established, but their source data are mostly social media images (e.g. Twitter), not egocentric. Some egocentric video datasets with wearer identity annotations (e.g. FPSI (Fathi et al., 2012), EVPR (Hoshen & Peleg, 2016), IITMD (Thapar et al., 2020a)) can be employed for wearer identification evaluation, but their potential is limited by the insufficient participants and scene diversity.

**Privacy Preservation in Egocentric Vision.** A straightforward solution is to disable the camera when sensitive information are detected (Templeman et al., 2014; Korayem et al., 2016). Beyond this, a line of work proposes to redact sensitive information in an egocentric video using processing techniques such as image degradation (Dimiccoli et al., 2018), object replacement (Hasan et al., 2017), and anonymization transformation (Thapar et al., 2021). Another line of work investigates how to perform utility tasks

with privacy-preserving representation of the egocentric videos/images (e.g. extremely downsampled video (Ryoo et al., 2017), text description (Qiu et al., 2023)) instead of the raw RGB data. Despite abundant research, they primarily focus on third-person subjects appearing in egocentric videos. Our work distinguishes itself from them by taking a new perspective, i.e. privacy concerns around the camera wearer.

**Egocentric Person Identification.** Person identification has been well-studied in third-person video settings but remains less explored in egocentric scenarios, where the subject can be either individuals in the camera's field of view or the camera wearer. For the former, the identification usually relies patterns of the face (Farringdon & Oni, 2000; Krishna et al., 2005; Mandal et al., 2014; Chakraborty et al., 2016) or body part (Fergnani et al., 2016). The identification of the wearer typically depends on head motion signature (Hoshen & Peleg, 2016; Thapar et al., 2020a), hand gesture (Thapar et al., 2020b; Tsutsui et al., 2021), and photographer style (Thomas & Kovashka, 2016). Some cross-view wearer identification approaches are proposed with additional third-person view (Yonetani et al., 2015; Poleg et al., 2015; Zhao et al., 2024) or top-view videos (Ardeshir & Borji, 2018b;a) as auxiliary data.

**Relationship Between Egocentric and Exocentric Videos.** The relationship between egocentric and exocentric videos has been investigated in applications such as knowledge transfer (Li et al., 2021), cross-view generation/translation (Liu et al., 2020; 2021; Luo et al., 2024b;c) and retrieval (Elfeki et al., 2018; Yu et al., 2020; Xu et al., 2024). The application of cross-view retrieval to the wearer privacy attack has yet to be thoroughly investigated.

## 3. Benchmarking Privacy in First-Person View

Most privacy-preserving vision addresses *third-person* video, equating privacy to (in)ability to recognize faces or other features that reveal personal information, like addresses or phone numbers. While this is concerning for egocentric videos, it fails to capture the full range of privacy risks posed by the latter, which can also expose information about the camera wearer's identity, demographics, and surroundings. To address this problem, we propose **EgoPrivacy**, a multidimensional privacy benchmark for egocentric vision.

### 3.1. Privacy Definition

We consider three types of privacy information and their potential of leakage in egocentric videos.

**Demographic privacy.** These attacks aim to recover demographic groups to which the camera wearer belongs. We

| Benchmark | Modality | #Subjects | #Scenes | Identity | Demographics | OOD Data |
|---|---|---|---|---|---|---|
| FPSI (Fathi et al., 2012) | Ego | 6 | ✗ | ✓ | ✗ | ✗ |
| EVPR (Hoshen & Peleg, 2016) | Ego | 32 | ✗ | ✓ | ✗ | ✗ |
| IITMD-WFP (Thapar et al., 2020a) | Ego | 31 | ✗ | ✓ | ✗ | ✗ |
| IITMD-WTP (Thapar et al., 2020a) | Ego+Exo | 12 | ✗ | ✓ | ✗ | ✗ |
| EgoPrivacy (Ours) | Ego+Exo | 731 | 123 | ✓ | ✓ | ✓ |

*Table 1.* Comparison of existing egocentric privacy benchmarks.

consider three such groups: gender, race, and age. While not fully identifying a person, these attributes can be leveraged to build user profiles for unwanted solicitation, e.g. targeted advertising, or discriminatory practices, e.g. misuse of race or gender information within health applications (Hoyle et al., 2015a; Price et al., 2017). Since they are categorical variables, we formulate demographic attacks as *classification* problems, where a predictor $f(\cdot)$ aims to infer a demographic attribute $a$ (e.g. *gender*, *race*, and *age*) of the camera wearer from egocentric video $\mathbf{x}$. This is illustrated in Figure 1. Privacy risk is measured by the demographic attribute classification accuracy

$$\text{Acc}(\mathcal{D}; f) = \frac{1}{|\mathcal{D}|} \sum_{(\mathbf{x}, a) \in \mathcal{D}} \mathbb{1}[f(\mathbf{x}) = a], \quad (1)$$

where $\mathbb{1}[\cdot]$ is the indicator function. Higher $\text{Acc}(\mathcal{D}; f)$ indicates that dataset $\mathcal{D}$ is more vulnerable to privacy attacks.

**Individual Privacy.** These attacks directly aim to recover the camera wearer *identity* $I$. As shown in Figure 1, this is formulated as a *retrieval problem*. A latent embedding is first learned, and a retrieval operation is performed to identify the nearest neighbors of the query $\mathbf{x}$. EgoPrivacy considers both the settings where the retrieved video is ego or exocentric. Privacy risk is measured by the *hit rate* at $k$ (HR@$k$) for retrieval of videos from the wearer of query $\mathbf{x}$

$$\text{HR@}k(\mathcal{D}; g) = \frac{1}{|\mathcal{D}|} \sum_{(\mathbf{x}, I) \in \mathcal{D}} \mathbb{1}[g^k(\mathbf{x}) \cap \mathcal{T}_I, \neq \emptyset] \quad (2)$$

where $g$ is the retrieval operator, $g^k(\mathbf{x})$ the top-$k$ retrieved videos and $\mathcal{T}_I$ the set of *videos of identity $I$* (the wearer) in dataset $\mathcal{D}$. Depending on the composition of the retrieval set $\mathcal{D}$, we further categorize the Individual Privacy into two tasks. If the retrieved videos are egocentric, the problem is formulated as ego-to-ego retrieval, where both the query $g^k(\mathbf{x})$ and the retrieval set $\mathcal{D}$ consist solely of egocentric videos. Conversely, if the retrieved videos are exocentric, the task becomes ego-to-exo retrieval, where given an egocentric query $g^k(\mathbf{x})$, the goal is to retrieve the exocentric videos from $\mathcal{D}$ with the same identity.

**Situational privacy.** Centering on situational awareness, these attacks aim to determine *where* or *when* an egocentric video clip was recorded. We consider two tasks: *scene* and *moment retrieval*. *Scene retrieval* is motivated by the fact

that because egocentric videos depict scenes similarly to exocentric videos, they have a similar risk of exposing private scene information (Chen et al., 2024). *scene retrieval* seeks to identify the location where the egocentric video was captured. Conversely, *moment retrieval* considers both, location (*where*) and the timing (*when*) of the footage, striving to pinpoint a precise moment in a corresponding exocentric clip, e.g. a clip captured by a different camera (Liu et al., 2024b; Luo et al., 2024a). As illustrated in Figure 1, both types of privacy are formulated as *retrieval problems* and evaluated with (2). *Scene retrieval* replaces $\mathcal{T}_I$ with $\mathcal{T}_S$, the set of video clips from $\mathcal{D}$ that are recorded in the scene of the query. For *moment retrieval*, $\mathcal{T}_I$ is replaced by $\mathcal{T}$, the set of exocentric video clips from $\mathcal{D}$ that are synchronized with the query video, e.g. footage from different third-person camera perspectives.

### 3.2. Benchmark Design

We provide a brief description of the EgoPrivacy benchmark here, further details on the datasets and annotation process can be found in Appendix A. EgoPrivacy is a benchmark of synchronized ego-exo video, built upon Ego-Exo4D (Grauman et al., 2024) and Charades-Ego (Sigurdsson et al., 2018a)[1]. It includes high-quality annotations for the three privacy categories discussed above: demographic labels (gender, age, and race) for each participant, as well as scene and identity annotations for each egocentric video clip. EgoPrivacy is composed of 5,625 video clips from Ego-Exo4D, captured by 839 diverse participants across 131 distinct scenes, and 4,000 clips of daily indoor activities from Charades-Ego, recorded by 112 participants in their homes.

All Ego-Exo4D and Charades-Ego clips include time-synchronized egocentric and exocentric videos along with identity annotations for each clip. However, demographic annotations are sparse since they are self-reported by camera wearers, and many were not collected. We leveraged the availability of exocentric videos to manually annotate the demographics of all participants. Camera wearer race, gender, and age labels were collected for all clips using Amazon Mechanical Turk. The label sets of the privacy classification problems were defined to reflect the make-up of the dataset. Gender classes

---

[1]All datasets used in the paper were solely downloaded and evaluated by UC San Diego.

are {*Female, Male*}[2], Race's are {*Asian, Black, White*}[3], Age's are {*Young, Middle-aged, Senior*}. For individual and situational privacy, we utilize the provided identity and scene annotations from the datasets. For moment retrieval, the location and timing labels are approximated based on clip footage, where each clip is treated as a distinct space-time instance.

The combination of videos from Ego-Exo and Charades-Ego facilitates the formulation of in-distribution (ID) and out-of-distribution (OOD) problem evaluations. Following the train/test split proposed in (Grauman et al., 2024), we split the Ego-Exo4D videos into a training set $\mathcal{D}_{train}$, that can be used for model finetuning, and a test set $\mathcal{D}_{test}$ for ID evaluation. Charades-Ego is then solely used as a test set for OOD evaluation.

Table 1 compares EgoPrivacy with previous egocentric privacy benchmarks (Fathi et al., 2012; Hoshen & Peleg, 2016; Thapar et al., 2020a), which are significantly smaller, focus solely on identity privacy, lack scene and demographic annotations, do not support OOD testing, and primarily consist of egocentric video data.

# 4. Egocentric Privacy Attack

In this section, we will propose our privacy attack to investigate the privacy concern of camera wearer in first-person views. We start by defining a set of threat models in Section 4.1 and then propose the attacker models in 4.2.

## 4.1. Attack Capability

We consider an adversary with the goal of obtaining one of the 7 types of privacy information of the camera wearer from an egocentric query video $\mathbf{x}$. We delineate a spectrum of capabilities ranging from minimal to extensive.

**Capability ① (*zero-shot*):** The adversary has no access to training data. This is the simplest class of attack, implementable by anyone with access to a foundation model.

**Capability ② (*fine-tuned*):** The adversary has access to a *labeled training dataset* $\mathcal{D}_{train}$ to fine-tune the model for attack purposes. $\mathcal{D}_{train}$ can include either egocentric videos, if $\mathcal{D}_{test}$ is egocentric, exocentric videos, if $\mathcal{D}_{test}$ is exocentric, or both in the case of moment and ego-to-exo identity retrieval.

**Capability ③ (*retrieval-augmented*):** The adversary has access to an identity labeled ego-exo paired training set (for ego-to-exo identity retriever) and an external pool of unlabeled exocentric videos $\mathcal{D}_{retr}$, which potentially includes the *identity* of the target egocentric query video $\mathbf{x}$.

---

[2]We note that these are perceived gender classes by the annotators

[3]Other racial categories were omitted due to the low representation in the dataset.

**Capability ④(*identity-level attack*):** In addition to the capabilities above, the adversary further ascertains whether two egocentric videos share the *same identity*, without necessarily identifying the individuals depicted.

We justify **Capability ③** and **Capability ④** in Appendix C, by outlining realistic threat scenarios in which they arise.

## 4.2. Implementation

In this section, we discuss the implementation of the threat models with different capabilities for each of the three privacy categories.

**Demographic Privacy** is modeled as a classification problem, as discussed in Section 3.1. Here, the classifier $f(\cdot)$ is implemented with a multi-modal foundation model. Capability ①: $f(\cdot)$ is applied to $\mathcal{D}_{test}$ in a zero-shot manner. Capability ②:$f(\cdot)$ is finetuned on $\mathcal{D}_{train}$ and tested on $\mathcal{D}_{test}$. We consider the *in-distribution* (ID), i.e. both $\mathcal{D}_{train}$ and $\mathcal{D}_{test}$ are from Ego-Exo4D and the *out-of-distribution* (OOD) where $\mathcal{D}_{train}$ are from Ego-Exo4D and $\mathcal{D}_{test}$ from Charades-Ego. For the combination of capability ① / ② and the additional ③, both query $\mathbf{x}$ and retrieval dataset $\mathcal{D}_{retr}$ are fed to the identity retriever to obtain feature vectors and RAA is performed, as discussed in Sec 5.

**Individual & Situational Privacy** are formulated as a retrieval problem, with a suitable embedding model. Both query $\mathbf{x}$ and videos in $\mathcal{D}_{test}$ are mapped into the embedding to create feature vectors and those from $\mathcal{D}_{test}$ ranked by similarity to $\mathbf{x}$, using the cosine similarity metric. Capability ① is implemented by the embedding of the foundation model directly in a zero-shot manner. Capability ②: the embedding is fine-tuned on $\mathcal{D}_{train}$, as discussed in Sec 5.1. The capability ③ is only for demographic privacy and is thus omitted here.

# 5. Retrieval-Augmented Attack

We present a deeper dive into ego-to-exo retrieval under a novel *retrieval-augmented* attack, to highlight its potential to boost the efficacy of classification-based attack models.

## 5.1. Ego-exo Embedding

To perform ego-to-exo retrieval, a joint embedding space of ego and exo video clips is required. We follow recent progress on cross-modal metric learning (Morgado et al., 2021; Radford et al., 2021) and perform the ego-to-exo retrieval with an embedding learned by *contrastive learning* (Oord et al., 2018). A pair of egocentric $\mathbf{x}_i^E$ and exocentric $\mathbf{x}_i^X$ examples is mapped into a pair of feature vectors $(\mathbf{z}_i^E, \mathbf{z}_i^X)$ using a joint embedding $(\mathbf{z}_i^E, \mathbf{z}_i^X) = (g(\mathbf{x}_i^E), g'(\mathbf{x}_i^X))$ where the mappings $g, g'$ are learned with a contrastive loss function. This uses ego-exo video pairs from the same person (demographic or individual privacy) or space-time (situational) as positive pairs.

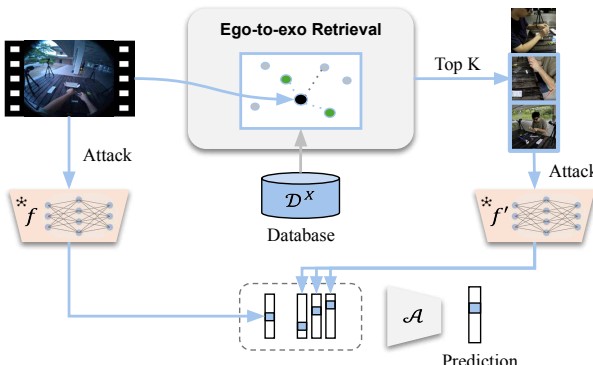

Figure 2. **Retrieval-Augmented Privacy Attacks.**

In general, several exocentric samples are associated with a single egocentric sample, either because the exocentric video is collected from multiple viewpoints or by definition of the retrieval task. For example, in individual privacy attacks all exocentric videos of the same camera wearer are considered successful retrievals, independently of whether they were shot at the same location or time. To account for this, we formulate the learning of the embedding as *supervised contrastive learning* (SupCon) (Khosla et al., 2020). This is a relaxed version of contrastive learning that distributes the loss evenly over all positive pairs

$$L(g, g') = -\sum_{i=1}^{N} \frac{1}{|P(i)|} \sum_{k \in P(i)} \log \frac{\exp(\langle \mathbf{z}_i^E, \mathbf{z}_k^X \rangle / \tau)}{\sum_{j \in N(i)} \exp(\langle \mathbf{z}_i^E, \mathbf{z}_j^X \rangle / \tau)},$$
(3)

where $P(i)$ is the set of exocentric feature vectors that are positive pairs of $\mathbf{z}_i^E$ and $N(i)$ a set of negative pairs. Sup-Con allows the unification of privacy types, individual and situational, simply by varying the definition of positive set $P(i)$. For individual privacy, $P(i)$ contains all exocentric examples $\mathbf{z}_i^X$ containing the camera wearer of $\mathbf{z}_i^E$. For situational privacy, $P(i)$ is restricted to the single exocentric video clip (single *take* in Ego-Exo4D) recorded in sync with $\mathbf{x}_i^E$. In both cases, the negative set $N(i)$ is formed by all other exocentric examples in the same minibatch as well as cached from past iterations of training.

### 5.2. Retrieval as Augmentation

Egocentric video inherently offers greater privacy protection for the subject compared to exocentric video, as faces and most of the body are obscured. However, if an adversary has access to the identity mapping between egocentric and exocentric videos, they can easily infer private information from the exocentric footage. We further notice that the ego-to-exo retrieval attack model as discussed in Section 4.2 performs this task exactly. Motivated by this, we propose *Retrieval Augmented Attack* (RAA) by exploiting an additional ego-to-exo retrieval model to retrieve exocentric videos for augmented prediction.

Formally, RAA is a two-stage privacy attack under the "retrieve, then predict" methodology, as illustrated in Figure 2. RAA assumes the availability of an external pool of *exocentric* data $\mathcal{D}^X$, which includes the individual behind the egocentric video. Given an egocentric query example $\mathbf{x}^E$, the attacker first uses an *ego-to-exo retrieval* module $g$ to rank all examples $\mathbf{x}_i' \in \mathcal{D}^X$ by their similarity to $\mathbf{x}^E$ in the embedding space $s_{g,g'}(\mathbf{x}^E, \mathbf{x}_i') = \langle g(\mathbf{x}^E), g'(\mathbf{x}_i') \rangle$; a support set $\{\mathbf{x}_{1:M}^X\} \subset \mathcal{D}^X$ is then formed by the top-$M$ most similar examples. The final output of RAA is the aggregation of the direct egocentric attack $f(\mathbf{x}^E)$ and the exocentric attacks on the retrieved examples $\{f'(\mathbf{x}_i^X)\}_{i=1}^{M}$:

$$f^{\text{RAA}}(\mathbf{x}^E, \{\mathbf{x}_{1:M}^X\}) = \mathcal{A}\left(f(\mathbf{x}^E), f'(\mathbf{x}_1^X), \ldots, f'(\mathbf{x}_M^X)\right)$$
(4)

where $f, f'$ are classification-based privacy attacks, such as gender predictors, on egocentric and exocentric inputs,[4] and $\mathcal{A}$ is an aggregation function that can be as simple as majority voting (hard voting) or weighted pooling (soft voting). By employing the simple voting ensemble, RAA without bells and whistles demonstrates significant effectiveness, improving the attack rate by a large margin.

## 6. Results

### 6.1. Experimental Setup

**Objectives.** We begin with a set of research questions and objectives of the experiments:

- Are egocentric videos a threat to the privacy of the camera wearer?
- To what extent do egocentric videos expose private information with different capabilities of the threat model?
- How effective is RAA in enhancing privacy attacks?
- What factors contribute to privacy vulnerabilities in egocentric videos?
- Do privacy attacks remain effective for out-of-distribution samples?

**Dataset.** All experiments are performed on the EgoPrivacy benchmark discussed in Section 3.2.

**Models & Baselines.** We consider a variety of models for launching the privacy attack, ranging from generalist vision-language models like CLIP (Radford et al., 2021; Fang et al., 2023) to video-centric models such as Video-MAE (Tong et al., 2022) and EgoVLPv2 (Pramanick et al., 2023) pre-trained on egocentric data, and large multimodel models (LMMs), such as LLaVA-1.5 (Liu et al., 2024a) and VideoLLaMA2 (Cheng et al., 2024b).

For exocentric demographic attacks, we also consider a straightforward face-based baseline, i.e. run face detection and demographic classification. Given the discovery that hand-based biometrics can be leveraged for inferring demo-

---

[4]One can use the same attack model for both views ($f = f'$).

| | OOD (Charades-Ego) | Capability ① | Capability ② | Gender Exo | Gender Ego | Gender RAA (+③) | Gender Δ | Race Exo | Race Ego | Race RAA (+③) | Race Δ | Age Exo | Age Ego | Age RAA (+③) | Age Δ |
|---|---|---|---|---|---|---|---|---|---|---|---|---|---|---|---|
| Random Chance | | | | | 50.00 | | - | | 33.33 | | - | | 33.33 | | - |
| Prior | | | | | 60.74 | | - | | 54.17 | | - | | 79.48 | | - |
| Hand-based | ✗ | N/A | | - | 45.33 | - | - | - | - | - | - | - | 65.30 | - | - |
| Face-based | ✗ | N/A | | 70.98 | - | - | - | - | - | - | - | 69.57 | - | - | - |
| CLIP_H/14 | ✗ | ✓ | ✗ | 78.64 | 57.89 | 67.35 | 9.46 | 60.04 | 45.21 | 60.98 | 15.77 | 73.51 | 72.02 | 76.23 | 4.21 |
| | ✗ | ✗ | ✓ | 96.03 | 85.04 | 90.74 | 5.70 | 88.57 | 85.82 | 88.00 | 2.18 | 69.28 | 58.5 | 63.89 | 5.39 |
| | ✓ | ✓ | ✗ | 89.80 | 70.00 | 77.31 | 7.31 | 60.14 | 46.09 | 59.42 | 13.33 | 48.02 | 20.75 | 26.42 | 5.67 |
| | ✓ | ✗ | ✓ | 83.56 | 56.74 | 62.44 | 5.70 | 87.39 | 73.22 | 77.01 | 3.79 | 29.90 | 29.70 | 29.92 | 0.22 |
| EgoVLP v2 | ✗ | ✓ | ✗ | 76.97 | 63.18 | 67.11 | 3.93 | 64.85 | 57.14 | 64.29 | 7.15 | 52.25 | 47.88 | 49.67 | 1.79 |
| | ✗ | ✗ | ✓ | 92.03 | 83.90 | 87.75 | 3.85 | 86.86 | 85.95 | 86.57 | 0.62 | 64.05 | 55.12 | 58.33 | 3.21 |
| | ✓ | ✗ | ✓ | 76.97 | 51.84 | 59.42 | 7.78 | 76.97 | 69.77 | 73.03 | 3.26 | 46.15 | 34.17 | 40.08 | 5.91 |
| VideoMAE_B/14 | ✗ | ✗ | ✓ | 72.42 | 63.69 | 70.65 | 6.96 | 75.16 | 66.73 | 73.49 | 6.76 | 78.21 | 79.73 | 81.70 | 1.97 |
| | ✓ | ✗ | ✓ | 67.97 | 42.09 | 55.40 | 13.31 | 72.08 | 46.50 | 57.42 | 10.92 | 30.57 | 29.70 | 30.33 | 0.63 |
| VideoMAE_L/14 | ✗ | ✗ | ✓ | 87.14 | 63.87 | 78.95 | 16.08 | 74.36 | 70.10 | 72.65 | 2.55 | 77.15 | 79.73 | 79.73 | 0.00 |
| | ✓ | ✗ | ✓ | 80.67 | 54.63 | 68.44 | 13.81 | 72.37 | 46.02 | 57.42 | 11.40 | 29.90 | 29.70 | 29.92 | 0.22 |
| LLaVA-1.5_7B | ✗ | ✓ | ✗ | 91.52 | 66.90 | 77.16 | 10.26 | 60.06 | 57.34 | 57.52 | 0.18 | 79.29 | 79.46 | 79.55 | 0.09 |
| | ✓ | ✓ | ✗ | 90.42 | 71.59 | 75.60 | 4.01 | 71.10 | 48.95 | 59.32 | 10.37 | 50.33 | 35.07 | 47.26 | 12.19 |
| LLaVA-1.5_13B | ✗ | ✓ | ✗ | 90.37 | 65.45 | 78.55 | 13.10 | 66.64 | 62.81 | 69.33 | 6.52 | 78.55 | 69.33 | 72.56 | 3.23 |
| | ✓ | ✓ | ✗ | 88.38 | 62.37 | 72.61 | 10.24 | 70.48 | 46.42 | 59.32 | 12.90 | 51.56 | 37.44 | 47.56 | 10.12 |
| VideoLLaMA2_7B | ✗ | ✓ | ✗ | 85.34 | 77.64 | 82.05 | 4.41 | 67.00 | 58.57 | 64.29 | 5.72 | 53.36 | 44.93 | 47.39 | 2.46 |
| | ✓ | ✓ | ✗ | 90.69 | 71.31 | 76.16 | 4.85 | 75.56 | 62.39 | 68.48 | 6.09 | 64.99 | 57.11 | 59.03 | 1.92 |
| VideoLLaMA2_72B | ✗ | ✓ | ✗ | 90.52 | 67.81 | 80.63 | 12.82 | 72.31 | 69.14 | 69.43 | 0.29 | 52.51 | 46.57 | 46.08 | -0.49 |
| | ✓ | ✓ | ✗ | 92.25 | 73.74 | 77.89 | 4.15 | 76.71 | 66.58 | 69.04 | 2.46 | 55.88 | 32.93 | 45.23 | 12.30 |

*Table 2.* Results on **Demographic Privacy**. Accuracy is calculated on a *per-video* basis. Δ indicates the accuracy increase brought by RAA (③) over ① / ②.

graphics such as gender and race (Matkowski et al., 2019; Matkowski & Kong, 2020), we also employed a hand-based demographics classifier as a baseline for egocentric demographic attacks.

**Training.** We add to the top of the foundation models with one layer of MLP for classification (demographic privacy) and use its representation layer for retrieval (individual and situational privacy). All models are trained with $1 \times$A100 with a batch size of 8. We use a learning rate of 1e-5 and adopt the AdamW optimizer with cosine learning rate decay. The default number of frames for one video is 8.

### 6.2. Main Results

**Are egocentric videos a threat to the privacy of the camera wearer?** We answer this by comparing different models with chance-level (lower bound) and exocentric performance (upper bound). As per Tables 2 and 3, we can clearly observe that 1) despite some lower than exocentric performance, all attack models in Tables 2 are higher than random chance by a large margin (more than 15%) for both Demographic, Identity and Situational Privacy; 2) except for zero-shot models, all fine-tuned models in Table 3 achieve significantly higher results compared to chance-level performance. The unsatisfactory performance of the zero-shot retrieval model is attributed to the fact that some of these models have not been trained on egocentric videos before, and hence fail to construct a meaningful ego-view representation. These results suggest that the risk of privacy leakage is a significant concern in egocentric vision.

**To what extent do egocentric videos expose private information under different capabilities of the threat model?** We evaluate the attack performance under a threat model with different capabilities outlined in Section 4.1. First, us-

ing zero-shot foundation models (①), we observe a really high *demographic* attack accuracy in Table 2, as illustrated by the highest 73.15%, 65.36% and 79.64% for gender, race and age respectively. This leads to the conclusion that even with minimum capabilities, the adversary can still perform a successful attack with up to 80% success rate. However, zero-shot models perform significantly worse on *situational* and *identity* attacks (Table 3), leaving these two privacy protected against capability ①.

When equipped with a training dataset (②), race and age results can be further improved to 72.01% and 80.72%, and retrieval-based attacks reach the highest of 81.2%, 50.31%, 89.21% and 15.43% top-1 hit rate on ego-to-ego, ego-to-exo identity, scene and moment retrieval tasks respectively. This suggests that, with access to some training data, an adversary could further extract more private information about the camera wearer from egocentric videos, thereby posing an even greater threat to privacy.

**Effectiveness of RAA.** With the additional capability ③, adversary is now able to perform the RAA attack. We demonstrate the delta after and before applying the RAA in Table 2. We can see a consistent improvement over all the models across all three tasks, with some even surpassing the exocentric baseline (e.g. EgoVLP v2). The most significant improvement is observed with the VideoMAE model on the gender classification task, achieving an increase in accuracy of over 16%. This result has demonstrated the effectiveness of RAA in most scenarios. We also observe some minimal improvement cases. These cases can be attributed to the small gap between egocentric and exocentric performance, leading to a minimal increase. We believe this is reasonable, as the performance on exocentric is generally seen as the upper bound of an egocentric privacy attack.

| | ① | ② | Identity | | | | Situational | | | |
|---|---|---|---|---|---|---|---|---|---|---|
| | | | **Ego→Ego** | | **Ego→Exo** | | **Scene** | | **Moment** | |
| | | | HR@1 | HR@5 | HR@1 | HR@5 | HR@1 | HR@5 | HR@1 | HR@5 |
| Random Chance | N/A | | 0.57 | 2.87 | 0.57 | 2.87 | 8.83 | 30.66 | 0.09 | 0.45 |
| *ID (EgoExo4D testset)* | | | | | | | | | | |
| CLIP$_{H/14}$ | ✓ | ✗ | 0.92 | 1.10 | 0.89 | 1.07 | 24.98 | 29.07 | 1.78 | 7.94 |
| | ✗ | ✓ | 86.52 | 97.66 | 51.52 | 66.31 | 89.25 | 90.05 | 12.72 | 41.85 |
| EgoVLP v2 | ✓ | ✗ | 4.85 | 8.31 | 7.31 | 18.38 | 28.64 | 28.88 | 1.96 | 7.94 |
| | ✗ | ✓ | 84.65 | 96.69 | 44.18 | 57.26 | 89.25 | 89.78 | 11.56 | 37.10 |
| VideoMAE$_B$ | ✓ | ✗ | 0.49 | 1.35 | 0.68 | 1.02 | 14.32 | 16.37 | 0.09 | 0.71 |
| | ✗ | ✓ | 76.76 | 95.12 | 44.00 | 61.02 | 85.75 | 87.37 | 13.80 | 39.25 |
| VideoMAE$_L$ | ✓ | ✗ | 0.88 | 1.74 | 0.93 | 1.07 | 13.60 | 15.98 | 0.00 | 0.45 |
| | ✗ | ✓ | 78.52 | 95.70 | 42.74 | 57.97 | 88.62 | 89.61 | 17.74 | 47.13 |
| *OOD (Charades-Ego testset)* | | | | | | | | | | |
| CLIP$_{H/14}$ | ✓ | ✗ | 0.59 | 1.04 | 0.90 | 1.89 | N/A | | 1.49 | 6.53 |
| | ✗ | ✓ | 53.07 | 78.24 | 37.33 | 49.69 | | | 8.22 | 26.69 |
| EgoVLP v2 | ✓ | ✗ | 5.03 | 9.90 | 6.74 | 17.44 | N/A | | 1.77 | 6.53 |
| | ✗ | ✓ | 42.63 | 66.42 | 34.87 | 46.57 | | | 7.79 | 28.74 |
| VideoMAE$_B$ | ✓ | ✗ | 0.69 | 1.49 | 0.83 | 1.95 | N/A | | 0.42 | 0.99 |
| | ✗ | ✓ | 44.25 | 68.87 | 36.08 | 48.78 | | | 8.07 | 27.33 |
| VideoMAE$_L$ | ✓ | ✗ | 0.57 | 1.58 | 1.04 | 2.38 | | | 0.48 | 1.16 |
| | ✗ | ✓ | 47.55 | 73.11 | 36.51 | 50.09 | | | 9.62 | 31.68 |

*Table 3.* Results on **Identity** and **Situational Privacy**. The hit rate is calculated on a per-video basis. Scene retrieval results are omitted for *OOD (Charades-Ego test set)* due to the absence of ground-truth labels in *Charades-Ego* dataset.

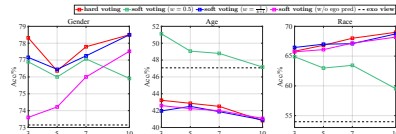

*Figure 3.* Performance of Retrieval Augmented Attack versus $k$.

We also notice that, even when the exocentric performance is lower than egocentric, RAA still offers improvements in some cases. We derive a hypothesis that RAA does not need the retrieval model to select the correct identity necessary to improve, but rather the retrieval model will cluster and group identities of similar attributes (of same gender, age and race, etc). To validate such a hypothesis, we conduct an experiment to see whether the ego-to-exo model groups identities of similar gender, race and age together. Specifically, we test how many top-1 and top-5 retrieved identities are of the same gender, age and race, as shown in Table 4. We can see that these retrieval models group people with similar gender, age and race together at a chance of over 82%, much higher than the chance it selects the correct identity (which is 50.31%). As long as the retrieval selects the identities with the correct demographic attributes, RAA can be improve the demographic classification.

| Gender | | Age | | Race | |
|---|---|---|---|---|---|
| Top-1 | Top-5 | Top-1 | Top-5 | Top-1 | Top-5 |
| 82.22 | 89.83 | 84.51 | 90.74 | 82.95 | 87.53 |

*Table 4.* Exo-to-ego identity retrieval as a demographic classifier.

**Ablation study on voting parameters.** As discussed in Section 5.2, RAA retrieves the top $k$ exocentric views to augment the egocentric view for prediction. Given these $k$ exocentric predictions and one egocentric prediction, an ensemble method is required to effectively combine them into a final output. In this Section, we explore two ensemble

| | | Gender | Race | Age |
|---|---|---|---|---|
| VideoLLaMA2$_{7B}$ | | 73.15 | 53.97 | _47.08_ |
| - w/ hard voting | | **78.32** | **65.82** | 43.23 |
| - w/ soft voting | $w = 0.5$ | 76.90 | 64.90 | **51.12** |
| | $w = 1/(k+1)$ | _77.16_ | _66.45_ | 41.97 |

*Table 5.* Different voting mechanisms for the Retrieval Augmented Attack. $w$: the weight over the egocentric prediction.

strategies and conduct ablation studies on various hyper-parameters. Hard voting, the simplest approach, involves voting on the predicted category and selecting the majority class. Given $k + 1$ predictions $f_1, \cdots, f_{k+1}$,

$$\hat{y} = \arg\max_{c \in \mathcal{Y}} \sum_{i=1}^{k+1} \mathbb{1}[f_i(x) = c].$$

We also consider weighted soft voting, where we weighted sum the predicted probabilities from the $k + 1$ views (softmax over logits) and use the category with the highest aggregated probability as the final prediction.

$$\hat{y} = \arg\max_{c \in \mathcal{Y}} \sum_{i=1}^{k+1} w_i f_i(x)$$

where $w_i$ is the weight for prediction from view $i$ As shown in Table 5, both hard and soft voting improve performance compared to the egocentric baselines. Hard voting generally yields better results for gender prediction, while soft voting consistently outperforms across all three demographic attributes. Therefore, we adopt soft voting as the default ensemble method. We further ablate the effect of the choice of $w$ in the soft voting ensemble, as shown in Table 5. Specifically, we compare two approaches: assigning evenly distributed weights ($w = \frac{1}{k+1}$) and assigning a weight of 0.5 to the egocentric prediction ($w = 0.5$). We also ablate the effect of the $k$ in top-$k$ retrieval in Figure 3, where $k = 3$

leads to the optimal performance for Gender and Age. For Race, we observe that a larger $k = 3$ leads to increasing performance.

**Can privacy attacks remain effective against out-of-distribution samples?** This question is practical, as privacy attacks often occur in real-world scenarios where in-distribution data is difficult to obtain. We use CharadesEgo as the OOD test set and evaluate all the attacker models described above, as presented in Table 2 and Table 3. We observe a consistent performance drop on the OOD data for all fine-tuned models, whereas the zero-shot foundation model maintains its original performance. This indicates a degree of overfitting during the fine-tuning stage and further underscores the privacy challenges inherent to egocentric videos: even with minimal attack capabilities (i.e., a zero-shot foundation model), an adversary can still launch effective attacks across varying data distributions.

**Capability ④** As discussed in Section 4.1, Capability ④ further assumes the ability of adversary to ascertain whether two egocentric videos share the same identity, therefore enabling it to ensemble the predictions over all the videos and infer the demographic attributes of the identity more effectively. We repeat the demographic privacy attacks of Table 2, but assume the additional Capability ④ of the adversary. We present the result in Appendix B due to limited space. Equipped with Capability ④, despite an improved performance on Gender egocentric and all exocentric videos, the performance drops on the rest of the tasks, surprisingly.

**What factors influence attacker models?** A preliminary comparison in Table 2 and Table 3 shows that EgoVLPv2 Fine-tuned consistently outperforms CLIP Fine-tuned, suggesting that temporal modeling aids adversaries in revealing private information. To investigate this effect, we evaluated models with MLP, Attention, and RNN layers atop the CLIP backbone, controlling for the number of parameters in each head. MLP layers map features to categories without temporal modeling, while Attention and RNN layers incorporate temporal information (temporal position embedding in Attention and recurrent nature of RNN). As shown in Figure 4: (1) Increasing the number of frames improves performance ($4 \Rightarrow 8$), but saturates beyond 8 or 16 frames; (2) Temporal modeling (Attention or RNN) consistently outperforms MLP. This effect is more pronounced. These findings are further validated for Identity and Situational Privacy in Appendix F.

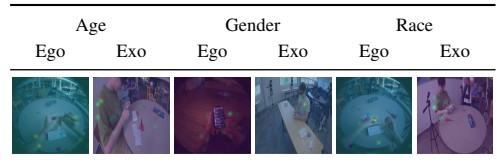

*Table 6.* Attention Visualization of LLaVa model.

**What leaks the privacy in the egocentric videos?** We vi-

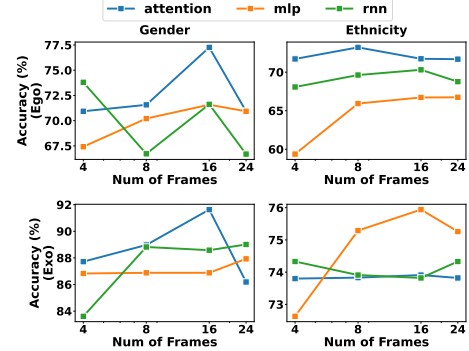

*Figure 4.* Performance of CLIP model with MLP, RNN, and attention head on Demographic Privacy.

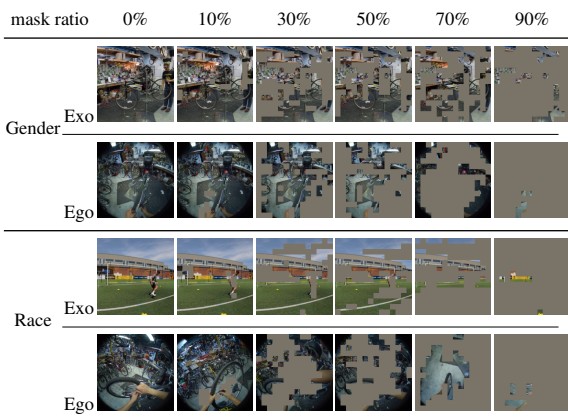

*Table 7.* Progressive masking of ego- and exo-video frames.

sualize the attention of LLaVA when it makes the prediction in Table 6. To further understand which patches contribute most to the prediction of privacy properties, we introduce a progressive masking method that incrementally masks the most important patches, as shown in Table 7. We refer to Appendix D for details of this method. Both visualizations reveal that significant attention is given to the wearer's hand or other biometric markers.

# 7. Conclusion

In this work, we introduced EgoPrivacy, a multidimensional benchmark of privacy in egocentric computer vision. By exploring demographic, individual, and situational privacy issues, we demonstrated that privacy information about the camera wearer can be extracted from first-person video data, even with off-the-shelf models in zero-shot. We proposed a retrieval-augmented attack, which further amplifies these threats by linking egocentric and exocentric footage of the same subjects. These results highlight the urgent need for privacy-preserving techniques in wearable cameras. We hope EgoPrivacy will drive future research on safeguarding privacy in egocentric vision while maintaining its utility.

## Acknowledgments

This work was partially funded by NSF awards IIS-2303153 and NAIRR-240300, the NVIDIA Academic grant, and a gift from Qualcomm. We also acknowledge the NRP Nautilus cluster, used for some of the experiments discussed above.

## Impact Statement

This research reveals a significant vulnerability in wearable camera systems, demonstrating that egocentric privacy attacks can be effectively executed even using readily available, unmodified models. Although the introduced privacy attack methods, such as *RAA*, are designed as red-teaming instruments aimed at enhancing privacy defenses, there exists a concerning potential for their misuse in unauthorized mass surveillance. Consequently, our findings highlight an urgent need for the development and implementation of robust privacy safeguards and proactive intervention mechanisms to mitigate risks associated with wearable technology. Furthermore, as EgoPrivacy builds upon Ego-Exo4D and Charades-Ego, it inherits their imbalances in geographic, gender, ethnic, and age representation, which raise concerns about the fairness problem. This emphasizes the need for future efforts to curate more equitable datasets in egocentric vision and privacy research, which will be the next step of our work.

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

# A. Dataset

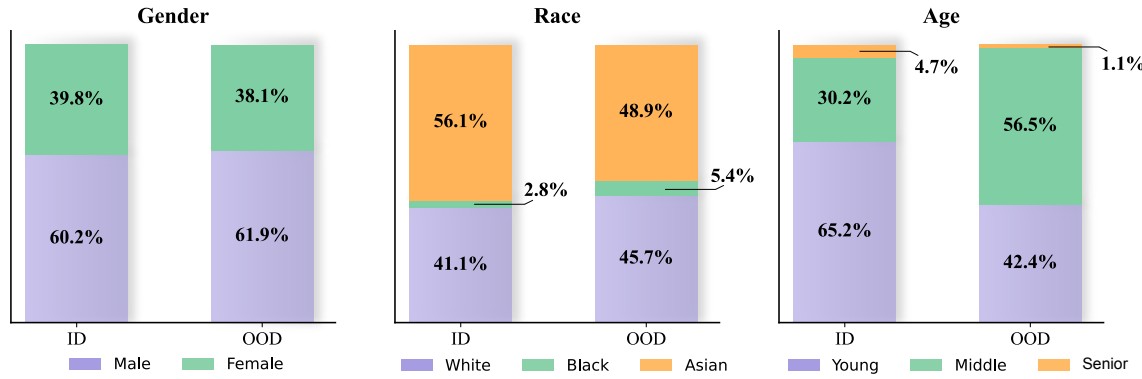

*Figure A.1.* Distributions of demographic labels in EgoPrivacy (ID). and EgoPrivacy (OOD).

**Data sources.** We build our EgoPrivacy upon two prior datasets with egocentric and exocentric annotation—Ego-Exo4D (Grauman et al., 2024) and Charades-Ego (Sigurdsson et al., 2018b). Ego-Exo4D comprises paired egocentric and exocentric videos capturing skilled activities performed by 740 participants across more than 100 distinct scenes in 13 cities worldwide. The dataset's diversity and extensive annotations enable privacy research at an unprecedented scale, making this study feasible for the first time. In Ego-Exo4D, each recording contains one or multiple trials ("takes") of an activity, with each take spanning 2.6 minutes on average. The dataset was released with labels of participant IDs associated with each video as well as self-reported demographics of some of the participants, making it an ideal candidate for studying privacy in egocentric vision. Ego-Exo4D dataset also provides redundant exocentric recordings, where each egocentric video is paired 4 exocentric view footage. Following the official dataset split, each participant is assigned exclusively to one of the train/val/test sets, preventing leakage of identity or demographic information in learning the attack models. The other dataset we adopt for EgoPrivacy is the Charades-Ego dataset. Charades-Ego is a dataset featuring 7,860 videos of daily indoor activities recorded from both third-person and first-person perspectives, comprising 68,536 temporal annotations across 157 action classes. Both videos possess paired egocentric and exocentric videos fulfilling the first requirement. To further satisfy the second requirement, we undergo an annotation process to label each identity of its gender, race and age. We note here that both the Ego-Exo4D and Charades-Ego dataset comes with identity labels. This is beneficial as it can reduce not only the annotation for identity but also the annotation cost of demographics for each video (since we can now annotate at the identity level).

**Annotation Process.** All videos and participant data used in this study come from publicly released datasets where participants consented to data collection. For participants who did not voluntarily disclose demographic information, we use crowd-sourced annotations of *perceived* attributes based on their video appearances. We employ Amazon Mechanical Turk for demographic annotation. For each identity, we display 3 to 4 (depending on the availability) *exocentric* videos to the annotator and request the annotator to answer three multi-choice questions regarding gender, race and age respectively. For each identity, we hire five Turker to annotate and filter any annotation with confidence less than 80%. These perceived demographics do not necessarily reflect individuals' self-identities. All collected data are used solely for academic research on privacy risks in egocentric vision, and we take measures to safeguard the confidentiality of participant information.

# B. Identity-level Privacy Attacks (Capability ④)

We repeat the *demographic* privacy attacks of Table 2, but assume the additional capability ④ of attackers, i.e. the ability to ascertain whether two egocentric videos share the same identity. We expect the attacker to further improve the attack performance with this extra information, which is the case for gender egocentric and all exocentric videos, as shown in Table B.1. However, the performance on egocentric age and race surprisingly drops.

# C. Justification of Threat Model Capabilities

We discuss capabilities ③ and ④ and justify their necessity by illustrating their relevance to real-world scenarios. For capability ③, consider a case where the target individual is a student who shares egocentric videos online, and an adversary

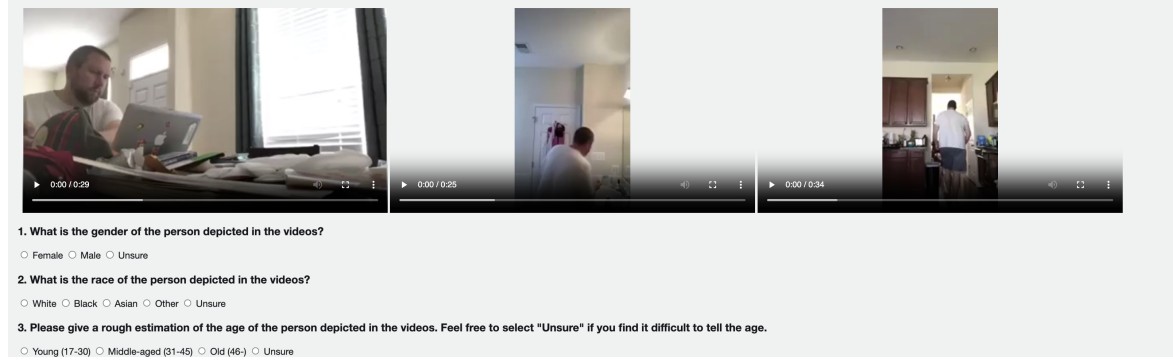

*Figure A.2.* Amazon Mechanical Turk web user interface for demographic annotation.

| | OOD (Charades-Ego) | Capability ① | ② | ④ | Gender Exo | Ego | RAA (+③) | Δ | race Exo | Ego | RAA (+③) | Δ | Age Exo | Ego | RAA (+③) | Δ |
|---|---|---|---|---|---|---|---|---|---|---|---|---|---|---|---|---|
| Random Chance | | | | | | 50.00 | | - | | 33.33 | | - | | 33.33 | | - |
| CLIP$_{H/14}$ | ✗ | ✓ | ✗ | ✓ | 84.97 | 62.07 | 71.26 | 9.19 | 62.84 | 59.17 | 62.13 | 2.96 | 73.03 | 67.63 | 73.99 | 6.36 |
| | ✗ | ✗ | ✓ | ✓ | 89.54 | 69.54 | 77.59 | 8.05 | 75.68 | 70.41 | 72.19 | 1.78 | 74.34 | 76.30 | 82.08 | 5.78 |
| | ✓ | ✓ | ✗ | ✓ | 93.02 | 76.19 | 79.43 | 3.24 | 68.60 | 58.33 | 63.71 | 5.38 | 54.65 | 20.24 | 27.00 | 6.76 |
| | ✓ | ✗ | ✓ | ✓ | 77.38 | 55.68 | 70.01 | 14.39 | 86.08 | 66.79 | 77.03 | 10.24 | 28.56 | 28.20 | 29.35 | 1.15 |
| EgoVLP v2 | ✗ | ✗ | ✓ | ✓ | 89.54 | 71.84 | 77.57 | 5.73 | 77.70 | 72.19 | 78.70 | 6.51 | 75.00 | 78.03 | 78.03 | 0.00 |
| | ✓ | ✗ | ✓ | ✓ | 78.16 | 55.32 | 68.02 | 12.70 | 77.32 | 61.77 | 73.54 | 11.77 | 29.20 | 28.20 | 28.57 | 0.37 |
| LLaVA-1.5$_{7B}$ | ✗ | ✓ | ✗ | ✓ | 96.08 | 71.26 | 72.99 | 1.73 | 67.57 | 52.66 | 66.27 | 13.61 | 79.61 | 76.30 | 77.46 | 1.16 |
| | ✓ | ✓ | ✗ | ✓ | 92.71 | 71.43 | 77.59 | 6.16 | 72.33 | 52.90 | 66.50 | 13.60 | 52.88 | 37.48 | 41.48 | 4.00 |
| LLaVA-1.5$_{13B}$ | ✗ | ✓ | ✗ | ✓ | 97.39 | 67.24 | 74.14 | 6.90 | 70.95 | 59.76 | 64.50 | 4.74 | 78.95 | 60.12 | 76.88 | 16.76 |
| | ✓ | ✓ | ✗ | ✓ | 95.35 | 71.43 | 78.56 | 7.13 | 70.24 | 52.69 | 62.42 | 9.73 | 52.88 | 36.72 | 42.38 | 5.66 |
| VideoLLaMA2$_{7B}$ | ✗ | ✓ | ✗ | ✓ | 98.04 | 77.01 | 80.46 | 3.45 | 77.03 | 60.36 | 74.56 | 14.20 | 56.58 | 42.77 | 52.60 | 9.83 |
| | ✓ | ✓ | ✗ | ✓ | 92.85 | 72.56 | 78.39 | 5.83 | 77.01 | 62.97 | 69.55 | 6.58 | 67.92 | 57.11 | 59.49 | 2.38 |
| VideoLLaMA2$_{72B}$ | ✗ | ✓ | ✗ | ✓ | 98.04 | 72.41 | 83.33 | 10.92 | 72.97 | 63.91 | 71.60 | 7.69 | 80.26 | 76.30 | 82.08 | 5.78 |
| | ✓ | ✓ | ✗ | ✓ | 95.33 | 74.54 | 79.90 | 5.36 | 77.92 | 68.22 | 70.35 | 2.13 | 57.01 | 33.88 | 47.09 | 13.32 |

*Table B.1.* Results on **Demographic Privacy**. Accuracy is calculated on a *per-identity* basis with the assumption of capability ④.

gains access to surveillance cameras in public areas of the student's school. Capability ④ is even more pervasive: here, the target posts multiple egocentric videos on social media, allowing an adversary to infer that all videos associated with the same account belong to a single individual. The objective of the adversary is then, given all the egocentric videos in the same account, infer the privacy attributes and information of the account owner. These examples highlight the practical relevance and necessity of these capabilities within our threat model.

## D. Details of Progressive Masking Method

In order to explore what features exactly in the video and frames that leaks the privacy information. We derive a progressive masking method that incrementally masks the most important patches. Specifically, we initialize a mask with values between 0 and 1 and perform gradient ascent on the mask with respect to the privacy property prediction loss. By gradually increasing the number of masked patches and employing early stopping once a predefined threshold is reached, we constrain the masking process to reveal the patches most critical to the model's decision.

## E. Biometric Classifier

For the hand-based model, we trained a ResNet50 classifier on the publicly available 11K Hands dataset (Afifi, 2019), which contains gender and age labels (but lacks race annotation). During inference, hand regions were first detected and cropped from egocentric video frames using a YOLO-based hand detection model (Cansik, 2020). The resulting hand crops were then passed to the trained ResNet50 classifier to predict demographic attributes. To aggregate predictions across multiple hand regions, we applied majority voting.

For the face-based model, we employed the FairFace model, pretrained on the FairFace dataset (Karkkainen & Joo, 2021), together with RetinaFace for robust face detection (Deng et al., 2019). Faces were detected and cropped from exocentric video frames using RetinaFace, after which the cropped images were input to the FairFace model to predict demographic attributes such as gender and age. As shown in the second section of Table 2, these biometric methods perform substantially worse than even the zero-shot foundation model, likely due to a pronounced distribution gap between the small, curated

datasets (hand/palm and face images) used for training and the more diverse, in-the-wild images in EgoPrivacy.

## F. Effect of Temporal Modeling in Identity and Situational Privacy

We validate the observation in Section 6.2 that temporal modeling is effective for adversary to reveal egocentric privacy, as shown in Figure F.1

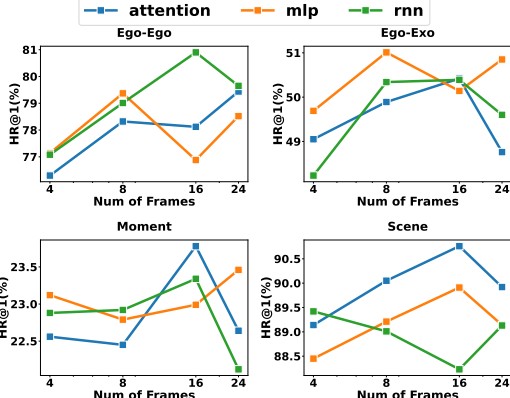

*Figure F.1.* Performance of Clip model with mlp, rnn and attention head on Identity and Situational Privacy.

