# OpenReview forum: "EgoPrivacy: What Your First-Person Camera Says About You?"
_ICML.cc/2025/Conference — ICML 2025 poster_

### Official Review · Reviewer_Nstm · 2025-03-10

**Overall Recommendation:** 3

**Summary:**

This paper presents an egocentric benchmark, termed EgoPrivacy, to analyze the potential information leakage in egocentric videos. The authors evaluate several vision-language models on EgoPrivacy to demonstrate that private information can be easily compromised by these models, highlighting the necessity of privacy-preserving techniques.

**Claims And Evidence:**

YES

**Essential References Not Discussed:**

This paper studies ego-to-exo cross-view retrieval, and propose a retrieval-augmented attack technique.

However, [1] also adopted a cross-view retrieval augmentation technique in egocentric vision, but it is not discussed in this paper.

[1] Retrieval-Augmented Egocentric Video Captioning. CVPR 24

**Experimental Designs Or Analyses:**

YES

**Methods And Evaluation Criteria:**

YES

**Other Comments Or Suggestions:**

There are several typos, e.g.
(1) Line 147, 'In this work' should be 'in this work';
(2) Line 253 'but as aa';
(3) Line 365, 'the moels';
(4) Line 437 'Appendix ??'.

**Other Strengths And Weaknesses:**

Advantages:
1. This paper provides insights into the implications of first-person camera data for personal privacy and security.

2. The proposed retrieval-augmented attack enhances the performance of vision-language models across different tasks.

Weaknesses:
1. The authors demonstrated the effect of retrieval augmentation in grouping similar gender, age, and race. In my opinion, key factors that affect the performance of retrieval augmentation also include: (1) the number of retrieved samples; (2) the retrieval mechanism; and (3) the method of integrating retrieved samples into the final prediction.

2. In Table 5, the comparisons are somewhat confusing. The authors utilized various models with different sizes and numbers of input frames. I do not see a fair comparison among these approaches. Additionally, in Line 409, the authors claimed that modeling temporal information helps reveal privacy information. I think this conclusion is not sufficiently solid, as CLIP only uses 8 frames while EgoVLPv2 uses 128 frames. It is unclear to me whether the improvement is due to the number of input frames or the model architecture.

3. The authors claimed that EgoPrivacy is a multimodal benchmark; however, only 4 to 5 vision-language models are included for comparison, which is insufficient.

4. Some related works regarding cross-view retrieval are missing [1,2,3,4]. Furthermore, it seems that the proposed Retrieval-Augmented Attack (RAA) shares a similar idea with [2], and I wonder what the differences are between them.

[1] Objectrelator: Enabling cross-view object relation understanding in ego-centric and exo-centric videos. Arxiv 24.

[2] Retrieval-Augmented Egocentric Video Captioning. CVPR 24

[3] EgoTransfer: Transferring Motion Across Egocentric and Exocentric Domains using Deep Neural Networks. Arxiv 16.

[4] First- And Third-Person Video Co-Analysis By Learning Spatial-Temporal Joint Attention. TPAMI 20.

**Questions For Authors:**

1. The authors adopted several vision-language models as baseline methods, including embedding-based models (CLIP and VideoMAE) and VLM-based models (LLaVA and VideoLLaMA2). It is straightforward to add a classification/retrieval head on top of the embedding-based models. However, regarding VLM-based models, it is not very clear to me where to place the classification/retrieval head and how zero-shot evaluation is conducted using VLMs. For example, did you use the output embedding from the language model? Did you input any language instructions?

2. As far as I know, EgoVLPv2 is trained using 4/16 frames, while the authors adopted 128 frames for evaluation. Are there any specific concerns regarding this? It would be fair to report the results using 4/16 frames, especially for zero-shot evaluation.

**Relation To Broader Scientific Literature:**

This paper provides insights into the implications of first-person camera data for personal privacy and security.

**Theoretical Claims:**

N/A

---

> ### Author Rebuttal · Authors · 2025-03-31
>
> ```>>> Q1``` Key factors that affect the performance of RAA.
> ```>>> A1``` Thanks. In fact, we do provide an analysis of these factors in Appendix B and Figure 6, due to limited space. We show that, overall, soft voting with 𝑤=0.5 yields the best performance, and varying the top‑k retrieved exo-predictions in RAA shows a performance plateau around k=3 or k=5 due to inaccurate additional exo-views, and including the ego prediction improves the attack.
>
> ```>>> Q2``` Comparisons with different input frames are not fair.
> ```>>> A2``` Thanks. We agree that comparing with the frame number is not entirely fair. Our original intention was not to make direct comparisons, but rather to showcase the best attack performance of each model within the same computational constraints (8xRTX 4090 for training). We here provide a more equitable comparison by using the same number of frames across all models.
> |               | Variant | Frames | identity | | | |situational ||        |       |
> |---------------|--------|--------|--------|-------|-------|-------|-------|--------|-------|-------|
> |               |         |        |EgoEgo     || EgoExo||Scene||Moment|
> |               |         |        |HR@1   |HR@5   |HR@1|HR@5|HR@1|HR@5|HR@1|HR@5|
> | CLIP (DFN)    | ViT-H/14|  8     |79.37|96.97|49.69|63.51|89.21|89.56|13.21|39.57|
> | EgoVLP v2     | 7B      |  8     |73.49|90.86|44.58|57.32|78.95|85.28|2.23|7.05|
> | VideoMAE      | ViT-B/14|  8     |54.41|77.47|22.78|31.52|57.47|63.84|6.97|20.33|
>
> Although EgoVLPv2 performs worse than CLIP with 8 frames, it offers better computational efficiency due to its early fusion, allowing for improved performance with more frames, thus achieving the best attack results while balancing performance and memory consumption. We will also discuss this point in response to Q7. We will add theresults to Table 3.
>
> Regarding temporal modeling, we acknowledge it is not sufficiently rigorous. As noted in Section 6.2, we position this as only a preliminary observation due to the limited time and space. We present a more thorough investigation by applying RNN, Attention (with pos emb) and MLP of similar param size on top of CLIP (see https://imgur.com/a/YMXXZMx). We show that both gender and race attack benefits from temporal modeling, while age is the exception.
>
> ```>>> Q3``` Only 4 to 5 vision-language models are included for comparison, which is insufficient.
> ```>>> A3``` Thanks for the comments. While we appreciate your concern, we believe that 4 to 5 models are sufficient for a comprehensive study in this context. To address your concern within the time constraint, we additionally add Qwen2.5-VL [1] results below:
>
> |               | Variant| Gender |        |        | Race   |       |       | Age    |       |       |
> |---------------|--------|--------|--------|--------|--------|-------|-------|--------|-------|-------|
> |               |        | Exo    | Ego    | RAA    | Exo    | Ego   | RAA   | Exo    | Ego   | RAA   |
> | Qwen2.5-VL    | 7B     | 82.00  | 86.50  | 87.50  | 74.00  | 70.00  | 71.50  | 58.00  | 60.50  | 60.50  |
>
> ```>>> Q4``` Missing related work [1,2,3,4]
>
> ```>>> A4``` While cross-view retrieval work [2, 4] exists, its application to egocentric demographic privacy is limited—[1] emphasizes semantic retrieval for action captioning, and EgoTransfer [3] and ObjectRelator focus on motion and object recognition, whereas our method targets instance-level retrieval for privacy attacks; we will include a detailed comparison in the final version.
>
> ```>>> Q5``` typos
> ```>>> A5```  Thanks! We will revise all typos in the final version of the paper.
>
> ```>>> Q6``` How is zero-shot eval employed in VLM-based models (LLaVA and VideoLLaMA2)?
> ```>>> A6``` We leverage the ability of VLM to output free-form texts and format the privacy attack as open-ended VQA.
> An example of the prompt:
>
> ---
>
> This is a video taken by a wearable camera. What is the gender of the wearer?
> A. Female
> B. Male
> Answer with the letter of the correct option.
>
> ---
>
> We use template matching to extract the answer, as all VLMs reliably follow instructions, making it sufficient for result extraction.
>
> ```>>> Q7``` EgoVLPv2 is trained using 4/16 frames, while the authors adopted 128 frames for evaluation.
> ```>>> A7``` Indeed, EgoVLPv2 we adopt only accepts 4 frames as input, but videos in the EgoPrivacy benchmark contain 1000+ frames. To address this, we split the 128-frame input into 32 chunks of 4 frames, processed independently by EgoVLPv2. The results are then aggregated via a classification/retrieval head. For zero-shot evaluation, we score and ensemble all 32 chunks' outputs to generate the prediction. We found 128 frames to offer the best balance of performance and efficiency. Following the suggestion, we perform the experiment as mentioned above with only 8 frames to provide a fair comparison. This resulted in a slight decrease in performance, which supports our choice of 128 frames for evaluation.

---

> > ### Comment · Reviewer_Nstm · 2025-04-04
> >
> > Thanks for the clarification. My concerns have been resolved.

---

> > > ### Author Response · Authors · 2025-04-04
> > >
> > > Thank you for taking the time to review our rebuttal and for your valuable feedback and suggestions in the review! We will revise the final version to incorporate the additional results accordingly.
> > >
> > > Please do not hesitate if you have further questions regarding the paper.
> > >
> > > Thanks,
> > > Authors of the paper

---

### Official Review · Reviewer_qJjA · 2025-03-13

**Overall Recommendation:** 3

**Summary:**

The paper proposes a new benchmark, EgoPrivacy, which focuses on the privacy issues of first-person view videos and quantifies the related privacy risks. The EgoPrivacy benchmark encompasses three types of privacy issues: Demographic privacy, Individual privacy, and Situational privacy. Moreover, based on the EgoPrivacy benchmark, the author proposes the Retrieval-Augmented Attack (RAA), a two-stage privacy attack of retrieval-prediction, and verifies the effectiveness of RAA and the privacy attack capabilities of off-the-shelf foundation models on the EgoPrivacy benchmark.

## update after rebuttal
I thank the authors for their reply. All my concerns have been addressed. I will keep my original rating.

**Claims And Evidence:**

Yes.

**Essential References Not Discussed:**

None.

**Experimental Designs Or Analyses:**

Yes.

**Methods And Evaluation Criteria:**

Yes.

**Other Comments Or Suggestions:**

1. There is a missing comma in Equation 4.
2. At the end of line 253, "but as aa video" should be "but as a video".
3. "Appendix ??" in line 438 needs to be corrected.
4. The parentheses in image (d) of Figure 1 need to be fixed.
5. The author defines three types of privacy: Demographic privacy, Individual privacy, and Situational privacy. Regarding whether there are other privacy issues in the EgoPrivacy benchmark, for example, the videos shot by users may contain some privacy-leaking information such as bank card numbers.
6. Figure 1 is difficult to understand. Most of the images are from the egocentric perspective, and there are a few from the exocentric views. It's unclear what information the images are trying to convey. It is recommended that the author polish Figure 1.

**Other Strengths And Weaknesses:**

None.

**Questions For Authors:**

None.

**Relation To Broader Scientific Literature:**

The contributions of this paper have promoted privacy protection in the first-person perspective.

**Theoretical Claims:**

None.

---

> ### Author Rebuttal · Authors · 2025-03-31
>
> First of all, we sincerely thank the reviewer for the valuable feedback and for acknowledging that our work haspromoted privacy protection in the first-person perspective. We will address the concern below.
>
> ```>>> Q1``` There is a missing comma in Equation 4. At the end of line 253, "but as aa video" should be "but as a video". "Appendix ??" in line 438 needs to be corrected. The parentheses in image (d) of Figure 1 need to be fixed.
> ```>>> A1``` Thanks for your comment. We will revise the paper and fix all typos in the final version.
>
> ```>>> Q2``` The author defines three types of privacy: Demographic privacy, Individual privacy, and Situational privacy. Regarding whether there are other privacy issues in the EgoPrivacy benchmark, for example, the videos shot by users may contain some privacy-leaking information such as bank card numbers.
> ```>>> A2``` Thanks for the insightful comments. This is absolutely a significant privacy concern. However, our paper focuses specifically on demographic, situational, and individual privacy, which are directly related to the person in the context of egocentric videos.  Thus, the mentioned privacy issue, such as bank card numbers in egocentric videos, falls out of the scope of this paper. However, we cite relevant papers [1] for the reviewer's reference and consider this as a potential and important direction for future research.
>
> [1] Raina, Nikhil, et al. "Egoblur: Responsible innovation in aria." arXiv preprint arXiv:2308.13093 (2023).
>
> ```>>> Q3``` Figure 1 is difficult to understand. Most of the images are from the egocentric perspective, and there are a few from the exocentric views. It's unclear what information the images are trying to convey. It is recommended that the author polish Figure 1.
> ```>>> A3``` Thanks for the comment. Figure 1 is indeed a bit misleading. The goal of this figure was to illustrate different types of privacy risks: (a)-(c) represent demographic privacy, (d)-(e) represent identity privacy, and (f)-(g) represent situational privacy, as defined in Section 3.1. All input examples are presented in the egocentric view since the primary focus of this work is to investigate egocentric privacy. For retrieval-based privacy tasks (identity and situational), we consider both ego-to-ego retrieval (where both the query and target are in the first-person view) and ego-to-exo retrieval (where the target is in the third-person view).
>
> To improve clarity, we will annotate the two perspectives in the figure and include a more detailed explanation in the introduction.
>
> Again, thank you for the valuable suggestions, and we hope that our rebuttal has addressed all the concerns proposed. Please do not hesitate if you have any further questions or concerns. We look forward to your response and valuable opinions!

---

> > ### Comment · Reviewer_qJjA · 2025-04-03
> >
> > I thank the authors for their reply. All my concerns have been addressed. I will keep my original rating.

---

> > > ### Author Response · Authors · 2025-04-04
> > >
> > > Thank you for your prompt response and for your valuable feedback in the review! We will revise the final version to incorporate the additional results accordingly.
> > >
> > > Please do not hesitate if you have further questions regarding the paper.
> > >
> > > Thanks,
> > > Authors of the paper

---

### Official Review · Reviewer_Laqt · 2025-03-16

**Overall Recommendation:** 4

**Summary:**

This paper addresses the privacy risks associated with egocentric video and introduces EgoPrivacy, a new benchmark for evaluating privacy in egocentric computer vision. The authors categorize privacy risks into three types: demographic, individual, and situational privacy. Their study demonstrates that private information about the camera wearer can be extracted, highlighting serious concerns. Additionally, the paper proposes a enhanced way of privacy attack called Retrieval-Augmented Attack (RAA), which leverages an external pool of exocentric data to enhance privacy retrieval from egocentric data, further intensifying privacy risks.

**Claims And Evidence:**

This paper claims that egocentric videos can leak private information about the camera wearer. To support this claim, the authors develop the EgoPrivacy benchmark and conduct a comprehensive evaluation, demonstrating that a threat attack model significantly outperforms a random guess model. This finding suggests that privacy risks are indeed present in egocentric vision.

**Essential References Not Discussed:**

N/A

**Experimental Designs Or Analyses:**

Yes, the authors have established a comprehensive experimental framework to compare the proposed privacy attack model against a random choice baseline. The results demonstrate that even without fine-tuning, off-the-shelf foundation models can already retrieve or classify the camera wearer and situational information with notable accuracy.

**Methods And Evaluation Criteria:**

Yes, this paper proposes a new benchmark, EgoPrivacy, and introduces three types of privacy attacks: demographic privacy, individual privacy, and situational privacy. It also defines the corresponding evaluation metrics for assessing these privacy risks.

**Other Comments Or Suggestions:**

N/A

**Other Strengths And Weaknesses:**

1. I appreciate the authors’ efforts in advancing the “attack” side of egocentric video privacy research. However, I would also like to see a short discussion on potential mitigation strategies to counteract this type of privacy attack, based on the insights gained from these experiments.
2. The notation of “1,” “2,” “3,” and “4” with circles in Section 4.1 is somewhat confusing. Could the authors clarify which models in Tables 2 and 3 correspond to these four threat models? A clearer way to distinguish and denote different types of models would improve readability.
3. It is interesting to observe in Table 2 that for Gender and Race, "Exo" consistently outperforms "Ego" before and after fine-tuning. However, for Age, after fine-tuning, "Exo" performs worse than "Ego". Could the authors provide insights into why this occurs?

**Questions For Authors:**

Please address my comments on the weakness section above.

**Relation To Broader Scientific Literature:**

The findings underscore the urgent need for privacy-preserving techniques in wearable cameras.

**Theoretical Claims:**

I reviewed the validity of the privacy attack claims in egocentric video. The experimental setups and results appear reasonable and correctly support the authors' conclusions.

---

> ### Author Rebuttal · Authors · 2025-04-01
>
> ```>>> Q1``` I would also like to see a short discussion on potential mitigation strategies to counteract this type of privacy attack, based on the insights gained from these experiments.
>
> ```>>> A1``` This is a great suggestion. While we intend to leave the study of privacy risk mitigation for future research, we will include additional discussion on the potential direction to solving this problem. First, we expect methods designed to protect third-person privacy (e.g., face blurring [1]) to be less effectve, as it does not address the privacy of camera wearer. Second, segmenting and obfuscating body parts (e.g., hand and feet) can reduce privacy risk, but also  makes utility tasks such as egocentric action recognition much harder. As such, an adaptive method for processing the egocentric vidoes is needed for the optimal trade-off between privacy and utility.
>
> In addition to the post-hoc techniques for software-level privacy preservation, another promising direction is to design cameras that can achieve hardware-level privacy-preservation, which further eliminate the privacy risk caused by the leak of raw camera measurements. The concept of privacy-preserving cameras has attracted attentions in the community [2,3,4], but their application in the egocentric setting remains to be investigated.
>
> [1]EgoBlur: Responsible Innovation in Aria. Arxiv 2308.13093.
> [2]Learning privacy-preserving optics for human pose estimation. ICCV'21.
> [3]Learning a Dynamic Privacy-Preserving Camera Robust to Inversion Attacks. ECCV'24.
>
> ```>>> Q2``` Could the authors clarify which models in Tables 2 and 3 correspond to these four threat models? A clearer way to distinguish and denote different types of models would improve readability.
>
> ```>>> A2``` Thanks for the question and the advice. We will first clarify the correspondence between threat models and the results in Table 2 and 3. Then, following your suggestion, we will introduce a new table that compares the performance across different threat models. This new table will replace part of Table 2 to enhance clarity, and we will split Table 2 into smaller, more focused tables in the camera ready version.
>
> To clarify:
>
> - **Zero-shot** results correspond to **Capability 1**. These results are shown in the first sections of Table 2 (excluding the RAA results). In Table 3, the rows marked with "ZS" correspond to the **Capability 1**.
>
> - **Fine-tuned** models belong to both **Capability 1** and **Capability 2**. Keep in mind that **Capability 1** is the basic capability, as all attacks require some form of query and model. In Table 2, the "Fine-tuned" section (second section) represents results related to **Capability 1 and 2** (excluding RAA results). In Table 3, these results are marked with "FT."
>
> - **RAA results**:
>
>     - In the **Zero-shot section** (first section) of Table 2, RAA corresponds to a composite of **Capabilities 1 and 3**, as it combines a base non-finetuned model with the ego-exo retriever and a pool of exo-videos.
>
>     - In the **Fine-tuned section** (second section) of Table 2, RAA is a composite of **Capabilities 1, 2, and 3**, since it involves fine-tuning in addition to the ego-exo retriever and exo-video pool.
>
> We hope this clarifies the relationship between threat models and their corresponding results in the tables.
>
> Here, we present the new table directly compares all the threat models. Due to space limit, only the results for the CLIP model is displayed here.
>
> |            | 1    | 2 | 3    | Model | Gender    | Race | Age   |
> |---------------|--------|--------|-------|-------|--------|-------|-------|
> | **ID (EgoExo4D)** | ✔️   | ❌ | ❌    | CLIP$_{H/14}$ | 57.89    | 45.21 | 72.02 |
> | | ✔️   | ❌ |✔️    | CLIP$_{H/14}$ |   67.35    |  60.98   |    76.23      |
> | | ✔️   | ✔️ |❌    | CLIP$_{H/14}$ |   68.87    |  70.92    |  79.73  |
> | | ✔️   | ✔️ |✔️    | CLIP$_{H/14}$ |    76.98   |   71.92  |     79.73     |
> |**OOD (CharadesEgo)** | ✔️   | ❌ | ❌    | CLIP$_{H/14}$ |  70.00   | 46.09 | 20.75 |
> | | ✔️   | ❌ |✔️    | CLIP$_{H/14}$ |77.31|  59.42   |    26.42      |
> | | ✔️   | ✔️ |❌    | CLIP$_{H/14}$ |   54.70    |   63.68   |  29.70  |
> | | ✔️   | ✔️ |✔️    | CLIP$_{H/14}$ |    69.65   |   74.09  |    29.92      |
>
>
> ```>>> Q3``` In Table 2 that for Gender and Race, "Exo" consistently outperforms "Ego" before and after fine-tuning. However, for Age, after fine-tuning, "Exo" performs worse than "Ego". Could the authors provide insights into why this occurs?
>
> ```>>> A3``` We believe this occurs due to the inherent subjectivity and complexity of age recognition. Unlike gender and race, which often have more distinct visual cues, age estimation is influenced by various factors such as lighting, facial expressions, and individual aging patterns. The fine-tuning process may have led the model to overfit to certain biases present in the egocentric data, resulting in better performance for ‘Ego’ compared to ‘Exo’ after fine-tuning.

---

### Official Review · Reviewer_i91u · 2025-03-17

**Overall Recommendation:** 3

**Summary:**

This paper examines the privacy implications of first-person (egocentric) video data, highlighting how demographic, individual, and situational information (e.g., age, identity, or time/location) may be inferred by combining egocentric footage with external exocentric (third-person) data.

The authors propose a new benchmark, “EgoPrivacy,” to systematically measure privacy leakage across various tasks, and introduce a method called Retrieval-Augmented Attack (RAA). In essence, they learn an embedding for cross-view retrieval:
$$
s\bigl(x^E, x^X\bigr) = \langle g(x^E), g'(x^X)\rangle,
$$
where $x^E$ is an egocentric video clip, $x^X$ is an exocentric clip, and $g, g'$ are learned transformations.

By finding exocentric clips that match an egocentric query, the method can enhance classification-based inferences (e.g., recognizing a camera wearer’s demographic attributes) through an aggregator that fuses outputs of both egocentric and exocentric predictions. The paper reports improved accuracy and retrieval hit rates on tasks such as demographic classification and identity/scene retrieval, arguing that egocentric data can expose more sensitive information than expected.

**Claims And Evidence:**

The main claim is that first-person video alone reveals sensitive information about the camera wearer—specifically demographics (age, gender, race), unique identity (via a retrieval task), and situational context (scene or moment in time). Evidence is shown by comparing baseline classifiers (or retrieval models) to the new RAA approach on tasks like:

1. **Demographic Privacy**: predicting a wearer’s attributes from short egocentric clips.
2. **Individual Privacy**: retrieving the same user’s other egocentric or exocentric clips.
3. **Situational Privacy**: identifying which place or time segment a clip belongs to by matching with exocentric footage.

Reported numbers exceed random-chance baselines. In classification tasks, accuracy improves once the external exocentric data is leveraged via:
$$
f^{\mathrm{RAA}}\bigl(x^E\bigr) = \mathcal{A}\Bigl(
   f \bigl(x^E\bigr),
   \{ f'(x^X)\}_{x^X  \in \text{top-}k \text{retrievals}}
\Bigr).
$$
The authors display performance gains when the aggregator combines predictions from both egocentric and exocentric sources. These empirical findings appear coherent with their overall contention that first-person data, especially when matched to exocentric footage, can substantially reveal private information.

**Essential References Not Discussed:**

No specific omitted references stand out as critical. The authors cite a variety of prior works about egocentric data and privacy, as well as multi-modal retrieval strategies. Most fundamental ideas from cross-view embedding and large-scale pretrained vision models are mentioned.

**Experimental Designs Or Analyses:**

The paper’s experiments cover zero-shot usage of large pretrained models (e.g., CLIP) versus fine-tuned models, and finally show that supplementing with cross-view retrieval (RAA) can further heighten the privacy leakage metrics. The evaluation on multiple tasks (demographic classification, identity retrieval, location/time retrieval) is displayed with tables comparing results from these different approaches.

The design is clear, and the tables illustrate how the RAA method surpasses baseline performance. The authors also compare results across different model architectures (Vision Transformers, specialized egocentric models, etc.). That said, there is little discussion of possible embeddings that fail to find meaningful matches, or of whether the aggregator function might occasionally be misled by false matches (though the final numbers still reflect improved average success).

**Methods And Evaluation Criteria:**

The work sets up multiple classification and retrieval tasks on the authors’ new benchmark, EgoPrivacy, which combines labeled egocentric-exocentric pairs from different sources. Classification accuracy and retrieval hit rates (HR@k) are the key metrics for privacy “attacks”.

Defining classification-based tasks for demographics, along with retrieval-based tasks for identity or situational context, matches the privacy categories described. The aggregator function itself is not elaborated in detail regarding weighting or potential variations, but the main idea follows standard voting or ensemble approaches and is shown to yield a quantitative boost in identifying private attributes.

**Other Comments Or Suggestions:**

I have no additional remarks regarding clarity or style. The overall presentation is straightforward, and the major claims are supported by numerical results.
Typographical Note: There appears to be a minor typographical error at line 437 in the main text (“Apeendix??”).

**Other Strengths And Weaknesses:**

- **Strength**: The paper addresses a focused set of privacy tasks with consistent metrics, demonstrating that even off-the-shelf large pretrained models—when paired with cross-view retrieval—can significantly reveal personal attributes from seemingly “private” first-person footage. This underscores the potential privacy threat in a straightforward manner.
- **Weakness**: The authors do not delve into an analytical framework for explaining how often retrieval collisions or false matches might occur in large datasets. One could write a probability-based expression for unintended collisions
  $$
  \mathrm{Pr}\bigl(\langle g(x^E), g'(x^X)\rangle > \delta\bigr)
  $$
  in the presence of many extraneous $x^X$ samples, but the paper remains mostly empirical. Nevertheless, the reported experiments indicate robust performance in the tested scenarios.

**Questions For Authors:**

1. **Exact Aggregation Mechanism**
   - How precisely is the aggregator $\mathcal{A}$ combining predictions from $f(x^E)$ and those from the retrieved exocentric clips $\{f'(x_i^X)\}$? Is it a simple average, or are there confidence-based weights?

2. **False-Match Considerations**
   - Do you have any observations on how often RAA might retrieve irrelevant exocentric videos when the embedding space has many visually similar but semantically mismatched samples?

**Relation To Broader Scientific Literature:**

The paper draws on established techniques of cross-view retrieval, person re-identification, and multi-modal vision-language modeling. It merges these techniques to focus on wearer-centric privacy leakage. The authors refer to prior references dealing with first-person privacy concerns and cross-modal retrieval, indicating that they build on known frameworks while extending them in the context of privacy threat modeling.

**Theoretical Claims:**

There are no extensive derivations or formal proofs regarding the retrieval process or classification bounds. The authors primarily rely on empirical demonstrations and do not present additional materials that formally analyze potential retrieval mismatches or performance guarantees.

Without more formal exploration of retrieval accuracy, one might question how robust the method is to large-scale collisions in the space of embeddings:
$$
\langle g(x^E), g'(x^X)\rangle >\delta.
$$
The paper does not provide a separate proof or theoretical section addressing these concerns. However, the experimental evidence still suggests that in practice, the method succeeds at matching relevant exocentric clips for the tasks at hand.

---

> ### Author Rebuttal · Authors · 2025-04-01
>
> ```>>> Q1``` Theoretical guarantees or formal proofs of retrieval Bound
>
> ```>>> A1```
> Thanks for the insightful comment. This paper indeed focuses more on empirical evidence of egocentric privacy risks, where measured by *hit rate* among top $K$ retrievals. This means we do *not* predefine thresholds for matching videos of the same identity, but use the k-th largest score among the candidate examples as an adaptive threshold. This implies a larger threshold as the number of idendidites in the test set increase, and hence a lower hit rate (equivalently, higher probability of false matches).
>
> Concretely, assuming a retrieval setting with $N$ individuals and $M$ candidate examples per identity, the chance-level hit rate @ $K$ is given by
> $$\textrm{HR}_\textrm{chance}@K = 1 - \frac{\binom{M(N-1)}{K}}{\binom{MN-1}{K}}.$$
> This approaches $1 - (1 - \frac{1}{N})^K$ when $M$ is large and $\frac{K}{N}$ when $N \gg K$, as is usually the case for large-scale datasets.
>
>
> ```>>> Q2``` How do false matches (collisions) impact method performance after aggregation? False-Match Considerations: Do you have any observations on how often RAA might retrieve irrelevant exocentric videos when the embedding space has many visually similar but semantically mismatched samples?
>
> ```>>> A2``` Great question! Yes, it is entirely possible that RAA may be misled by false positives in retrieval. The retriever used in RAA is based on the ego-exo retrieval model from EgoVLP v2. As shown in Table 3, this model achieves a 50.31% HR@1 and 66.82% HR@5. However, we empirically find that this relatively low retrieval rate does not significantly impact the demographic prediction accuracy, as discussed in the second paragraph of the "Effectiveness of RAA" section (Section 6.2).
>
> Despite the lower retrieval accuracy, RAA still enhances the attack's effectiveness. We hypothesize in L376-427 that even with imperfect retrieval, the process helps cluster and group identities with similar demographics (such as gender, age, and race) closer in its learned embedding space. This is evidenced in Table 4, where, despite a 50% retrieval accuracy, 82.22%, 84.51%, and 82.95% of the top-1 retrieved exo-videos share the same gender, age, and race as the ego-video, respectively. This suggests that even negative retrievals can *benefit* RAA if they are semantically close to the query.
>
>
> ```>>> Q3``` Minor typographical error at line 437 in the main text.
>
> ```>>> A3``` Thank you for pointing this out. We will revise the paper and fix this typographical error.
>
> ```>>> Q4``` What is the aggregation mechanism?
>
> ```>>> A4``` Thanks for this good question. In fact, we explore various aggregation mechanisms in Appendix B and Figure 6. The default mechanism involves a weighted sum of both the ego and exo predictions (referred to as soft voting), where the ego weight is set to 0.5, and the remaining 0.5 is evenly distributed across the exo predictions. In addition to this, we also examine hard voting (direct majority voting) and soft voting with equal weights assigned to all predictions. Our experiments show that, overall, soft voting with
> 𝑤=0.5 yields the best performance, improving all three demographic tasks consistently compared to the ego-only baseline (denoted by the dashed line).
>
> Additionally, we investigate the impact of varying the number of top-k retrieved exo-predictions used in RAA. Our findings reveal that there is a key pivot point around
> 𝑘=3 or 𝑘=5, beyond which increasing k does not improve performance. This is because additional retrieved exo-views often introduce noise due to their potential inaccuracy. We also perform an ablation study on the role of the ego prediction in RAA, showing that including the ego prediction generally enhances the attack's effectiveness. We also experiment with using the retrieved similarity as a confidence measure to weigh each prediction, however, this empirically fial to improv over even the majority voting.

---

> > ### Comment · Reviewer_i91u · 2025-04-04
> >
> > I appreciate the helpful clarifications, which resolved my concerns. Therefore, I will maintain the original score.

---

> > > ### Author Response · Authors · 2025-04-04
> > >
> > > Thank you for taking the time to review our rebuttal and for your valuable feedback and suggestions in the review! We will revise the final version to incorporate rebuttal accordingly.
> > >
> > > Please do not hesitate if you have further questions regarding the paper.
> > >
> > > Thanks,
> > > Authors of the paper

---

### Official Review · Reviewer_iyPv · 2025-03-22

**Overall Recommendation:** 1

**Summary:**

This paper investigates the privacy risks associated with egocentric videos. In particular, the authors introduce EgoPrivacy, a large-scale benchmark to evaluate privacy vulnerabilities across three axes: demographic privacy (gender, race, age), individual privacy (identity re-identification), and situational privacy (time and location). They further propose a novel Retrieval-Augmented Attack method, which boosts demographic attribute inference by retrieving relevant exocentric video footage associated with the egocentric clip. The paper evaluates zero-shot and fine-tuned models (CLIP, LLaVA, VideoMAE, and EgoVLPv2) on the proposed benchmark. The benchmark and findings are intended as red-teaming tools to inform and stimulate future privacy-preserving methods for egocentric vision.

**Claims And Evidence:**

**EgoPrivacy Benchmark**

The paper introduces EgoPrivacy, the first large-scale benchmark explicitly designed to evaluate privacy risks in egocentric video across demographic, individual, and situational dimensions. This contribution is well-supported: EgoPrivacy is constructed from two large datasets—Ego-Exo4D and Charades-Ego—and annotated for demographic attributes (gender, race, age), individual identity, and situational context (location and moment). The benchmark includes seven tasks, framed as classification and retrieval problems, to comprehensively assess these privacy risks.

**Benchmark Evaluation**

The paper offers a thorough empirical evaluation of the EgoPrivacy benchmark, revealing notable privacy vulnerabilities in egocentric video data. This claim is moderately well-supported: Although experimental results do show that general-purpose foundation models can reliably infer demographic, identity, and situational attributes at rates well above chance, the reported performance likely underestimates real privacy risks, as SoTA models specialized for these tasks would likely achieve significantly higher accuracy.

**Retrieval-Augmented Attack**

The paper proposes a novel privacy attack that leverages ego-to-exo retrieval augmentation to enhance demographic inference. This contribution is well-supported: Augmenting predictions with exocentric views significantly improves demographic classification performance across multiple models.

**Essential References Not Discussed:**

References to hand-based biometric models and datasets are missing. See **Experimental Designs Or Analyses** section.

**Experimental Designs Or Analyses:**

The experimental evaluation of the benchmark, while suggestive, is incomplete. Although the authors effectively demonstrate demographic leakage using general-purpose foundation models (e.g., CLIP, LLaVA), the reported performance likely underestimates the actual privacy risks.

Notably, exocentric classification results (e.g., ~90% accuracy for gender, ~75% for race) fall significantly below the ceiling performance of task-specific state-of-the-art models, which routinely exceed 98% for gender and 90–95% for race and age (e.g., FairFace, DEX).

Likewise, for egocentric classification, the authors do not explore hand-based biometric models, despite their relevance for predicting demographic and identity attributes in the absence of facial visibility. Prior work in biometric vision has demonstrated that hands carry sufficient cues to support identity and demographic inference with high accuracy (Matkowski et al., 2019), (Matkowski et al., 2020)

Several publicly available datasets could be leveraged to develop stronger baseline models or better contextualize the benchmark's difficulty:

- **PolyU-IITD-v3:** ~12k hand images with identity and ethnicity labels
- **11K Hands:** ~11k hand images annotated with identity, age, gender, and ethnicity
- **NTU-PI-v1:** ~8k palm and dorsal hand images with identity, age, gender, and ethnicity labels
- **CASIA:** ~5.5k hand images labeled with identity

Incorporating hand-based SoTA models or at least referencing their performance as upper bounds would strengthen the benchmark's credibility and offer a clearer picture of the limits of wearer privacy in egocentric settings.

**Methods And Evaluation Criteria:**

**Benchmark**

The data collection strategy is appropriate and well-grounded. The authors utilize two large-scale egocentric datasets—Ego-Exo4D and Charades-Ego—as sources of raw video data. To support demographic analysis, they augment these datasets, where necessary, with additional annotations via Amazon Mechanical Turk. The suite of proposed privacy tasks is well-motivated and reflects real-world concerns: demographic attribute classification (gender, race, age), identity re-identification, and situational inference (time and location prediction). These tasks collectively provide a comprehensive framework for evaluating wearer-centric privacy risks in egocentric video.

**Retrieval-Augmented Attack (RAA)**

The threat model for the RAA is well-founded, as an attacker could plausibly exploit both egocentric and exocentric videos to carry out a privacy attack.

**Other Comments Or Suggestions:**

NA

**Other Strengths And Weaknesses:**

Aside from the demographic annotations added to Charades-Ego, the benchmark primarily repurposes existing datasets (Ego-Exo4D and Charades-Ego), and thus offers limited novelty in terms of data collection. The core insight—that identity and demographic information can be extracted from egocentric video—is not particularly surprising, especially given prior work showing that such attributes can be inferred from hand images alone. The proposed Retrieval-Augmented Attack (RAA) closely resembles earlier cross-view retrieval methods, such as those by Fan et al. (2017) and Ardeshir et al. (2018), and does not introduce significant algorithmic innovation. Finally, the benchmark evaluation is not thoroughly conducted: it primarily relies on general-purpose models for the privacy attacks, rather than leveraging state-of-the-art models specifically trained for demographic or identity recognition, which would offer a stronger and more realistic assessment of privacy risks.

**Questions For Authors:**

NA

**Relation To Broader Scientific Literature:**

The paper sits at the intersection of computer vision, privacy, and egocentric video analysis, and introduces the first large-scale benchmark specifically designed to evaluate wearer-centric privacy risks across both egocentric and exocentric perspectives.

**Theoretical Claims:**

The paper does not present any theoretical claims.

---

> ### Author Rebuttal · Authors · 2025-04-01
>
> ```>>> Q1``` Although the authors effectively demonstrate demographic leakage using general-purpose foundation models (e.g., CLIP, LLaVA), the reported performance likely underestimates the actual privacy risks.
> ```>>> A1```  Thanks for the suggestion. While models tailored to privacy attacks may have an advantage, there's a significant gap between datasets used for facial attribute prediction—typically high-quality, cropped facial images (e.g., CelebA, UTKFace, FairFace)—and EgoPrivacy, which uses raw, distant egocentric video (see [image](https://bashify.io/i/fQwScK) left). *We argue that SOTA models perform well in ideal conditions, but their metrics aren't directly comparable to results on EgoPrivacy.*
>
> We also agree that current vision-language foundation models, despite performing surprisingly well, aren’t optimized for egocentric privacy tasks. Their performance could improve with more egocentric training data. This motivates our benchmark: to drive research, enable objective progress tracking, and advance attack and defense methods in the egocentric privacy domain.
>
> ```>>> Q2``` For egocentric classification, the authors do not explore hand-based biometric models.
> ```>>> A2``` Just as exocentric video frames with faces do not resemble images from high-quality facial attribute datasets (see A1 above), the distribution gap between hand/palmprint identification data and egocentric videos is even more dramatic, as shown in [image](https://bashify.io/i/fQwScK) (right). The vast majority of egocentric video frames do not show clear pictures of the hands of the camera wearer; even those that do tend to show the back of hands from first-person view, which contains far less identifiable information than the palm. This makes it very challenging to apply hand biometric methods to egocentric privacy tasks directly without nontrivial modifications or retraining. We will include a thorough discussion of specialized approaches to egocentric privacy attacks in the final version of the paper, as well as empirical comparisons to hand-based baselines.
>
> Regarding additional baselines in exocentric (face-based) and egocentric (hand-based), we have invested effort to reproduce the results of the suggested papers on our benchmark. However, due to the limited time and computation, as well as the unavailability of pretrained models in public and the scale of our evaluation set (over 5000 exo- and ego videos), we are unable to offer results at the moment, but hopeful can share results during the discussion phase and final version of the paper.
>
> ```>>> Q3``` Limited novelty in terms of data collection.
> ```>>> A3``` While we do repurpose Ego-Exo4D and Charades-Ego dataset for egocentric privacy, additional effort has been made on the annotation of demographic labels, on an unprecedented scale, which is a nontrivial task. This annotation further facilitates the significant research on the egocentric privacy that has received limited attention in prior egocentric vision literature.
>
> ```>>> Q4``` The proposed Retrieval-Augmented Attack (RAA) closely resembles earlier cross-view retrieval methods, such as those by Fan et al. (2017) and Ardeshir et al. (2018), and does not introduce significant algorithmic innovation.
> ```>>> A4``` Thank you for your comment. As discussed in Section 2, "Related Work on Egocentric Person Identification," we acknowledge the contributions of earlier cross-view retrieval methods [1, 2, 3, 4]. However, previous work has largely overlooked the impact they have on the demographic privacy of camera users, i.e. how cross-view retrieval techniques might facilitate security breaches of egocentric demographic privacy, which is the primary focus of our study.
>
> For instance, [1, 2] focus on identifying first-person camera wearers in third-person videos, but they do not explore the potential of these methods in enhancing demographic attacks. Similarly, [3] investigates cross-view techniques for retrieving motion features, while [4] uses cross-view retrieval to improve action and motion captioning at the semantic level, neither of which are designed with privacy concerns in mind. In the final version of the paper, we will include a more detailed discussion of the novelty of RAA in relation to these prior works.
>
> [1] Fan, Chenyou, et al. "Identifying first-person camera wearers in third-person videos." Proceedings of the IEEE Conference on Computer Vision and Pattern Recognition. 2017.
> [2] Elfeki, Mohamed, et al. "From third person to first person: Dataset and baselines for synthesis and retrieval." arXiv preprint arXiv:1812.00104 (2018).
> [3] Ardeshir, Shervin, Krishna Regmi, and Ali Borji. "Egotransfer: Transferring motion across egocentric and exocentric domains using deep neural networks." arXiv preprint arXiv:1612.05836 (2016).
> [4] Xu, Jilan, et al. "Retrieval-augmented egocentric video captioning." Proceedings of the IEEE/CVF Conference on Computer Vision and Pattern Recognition. 2024.

---

### Official Review · Reviewer_DYc1 · 2025-03-25

**Overall Recommendation:** 4

**Summary:**

This paper introduces EgoPrivacy, a benchmark and study on the privacy risks associated with egocentric (first-person) videos, revealing that substantial personal information about the camera wearer—such as demographics (gender, race, age), identity, and location/time—can be inferred even when the wearer’s face is not visible. The authors propose a novel Retrieval-Augmented Attack (RAA) that boosts privacy attacks by retrieving matching exocentric (third-person) videos, significantly improving inference accuracy. Through experiments with both zero-shot and fine-tuned models (like CLIP, LLaVA, and EgoVLPv2), they show that even minimal attacker capabilities can lead to effective privacy breaches, raising serious concerns about the use of wearable cameras and highlighting the urgent need for privacy-preserving approaches in egocentric vision.

### update after rebuttal

I have raised my score.

**Claims And Evidence:**

All claims are well supported.

**Essential References Not Discussed:**

N/A

**Experimental Designs Or Analyses:**

### Calculation of Random Accuracy

The authors claim that the random accuracy for predicting race is 33.3%. However, the distribution of the demographic is unbalanced per the appendix.

**Methods And Evaluation Criteria:**

Yes.

**Other Comments Or Suggestions:**

Missing Appendix letter (line 437)

**Other Strengths And Weaknesses:**

N/A

**Questions For Authors:**

1- Can the authors clarify the issue with random accuracy and then update the results and discussion if necessary?


Overall, the work is novel and well motivated. I will update my score upon addressing the critical issue above.

**Relation To Broader Scientific Literature:**

The paper builds meaningfully on prior work in visual privacy and egocentric vision by addressing gaps left by earlier datasets and studies. While prior egocentric datasets like FPSI and EVPR were limited in scale and scope, this work introduces a large-scale, richly annotated benchmark (EgoPrivacy) that uniquely targets privacy risks faced by the camera wearer. It builds on prior work by contributing a structured taxonomy of privacy types—demographic, individual, and situational—that had not been comprehensively studied together. The proposed Retrieval-Augmented Attack also builds on cross-view retrieval work (e.g., Elfeki et al., 2018), but repurposes it to demonstrate novel privacy vulnerabilities.

**Theoretical Claims:**

N/A

---

> ### Author Rebuttal · Authors · 2025-04-01
>
> ```>>> Q1``` Calculation of random accuracy
>
> ```>>> A1``` Thank you for the insightful question. We present the prior accuracy
>
> |               | Variant | Gender |        |       | Race   |        |       | Age    |        |       |
> |---------------|---------|--------|-------|-------|--------|-------|-------|--------|-------|-------|
> |               |        | Exo    | Ego   | RAA   | Exo    | Ego   | RAA   | Exo    | Ego   | RAA   |
> | Random Chance |        | 50.00  |       |       | 33.33  |       |       | 33.33  |       |       |
> |  Priored      |       | 60.74      |       |       | 54.17      |       |       | 79.48      |       |       |
> | **Zero-shot (ID)**|        |       |       |        |       |       |        |       |       |
> | CLIP (DFN)    |ViT-H/14| 78.64  | 57.89  | 67.35  | 60.04  | 45.21  | 60.98  | 73.51  | 72.02  | 76.23  |
> | SigLIP       |S0400M/14| 84.01  | 57.63  | 70.56  | 65.46  | 54.97  | 66.27  | 72.90  | 68.25  | 78.03  |
> | LLaVA-1.5     | 7B     | 91.52  | 66.90  | 77.16  | 60.06  | 57.34  | 57.52  | 79.29  | 79.46  | 79.55  |
> | Video-LLaMA2  | 7B     | 90.96  | 73.15  | 79.48  | 71.53  | 53.97  | 69.10  | 52.99  | 47.08  | 56.14  |
> |               | 72B    | 91.59  | 70.03  | 78.41  | 69.25  | 65.36  | 67.82  | 82.46  | 79.64  | 81.62  |
> |**Fine-tuned (ID)**|    |        |        |        |        |        |            |        |        |
> | CLIP (DFN)    |ViT-H/14| 88.33  | 68.87  | 76.98  | 73.93  | 70.92  | 71.92  | 77.15  | 79.73  | 79.73  |
> | EgoVLP v2     | 7B     | 84.85  | 71.81  | 77.88  | 71.46  | 72.01  | 75.57  | 77.11  | 80.72  | 81.88  |
> | VideoMAE      |ViT-B/14| 72.42  | 63.69  | 70.65  | 75.16  | 66.73  | 73.49  | 78.21  | 79.73  | 81.70  |
> |               |ViT-T/14| 87.14  | 63.87  | 78.95  | 74.36  | 70.10  | 72.65  | 77.15  | 79.73  | 79.73  |
>
> However, imbalance also poses challenges during training. Thus, we also perform additional studies that merge / remove the extremely imbalance category in the age / race demongraphic attacks. Specifically, we merge the old with the middle age in age and remove the black in the race, as these two categories are extremely minority in the dataset. We present additional results below:
> |               | Variant | Race   |        |       | Age    |        |       |
> |---------------|--------|--------|-------|-------|--------|-------|-------|
> |               |        | Exo    | Ego   | RAA   | Exo    | Ego   | RAA   | Exo    | Ego   | RAA   |
> | Random Chance |        | 50.00  |       |       | 33.33  |       |       | 33.33  |       |       |
> | Priored      |        | 55.93      |       |       | 79.48      |       |       | -      |       |       |
> | **Zero-shot (ID)**|        |       |       |        |       |       |        |       |       |
> | LLaVA-1.5     | 7B     |61.93|58.60|68.22|80.10|80.47|81.56|
> | Video-LLaMA2  | 13B    |71.53|53.97|69.10|52.99|47.09|56.14|   |  |
> | **Fine-tuned (ID)**|     |       |       |        |       |       |
> | CLIP (DFN)    | ViT-H/14|69.06|67.91|69.37|76.04|79.73|79.91|
> | EgoVLP v2     | 7B     |64.90|65.36|69.28|76.03|78.65|79.91|
>
> In general, we observe that gender and race attacks outperform prior accuracy by large margins, while the improvement of age attacks is less significant due to heavy imbalance in age demographics. We leave for future work the collection of more balanced egocentric video datasets for an unbiased evaluation of demographic privacy.

---

### Decision · Program_Chairs · 2025-05-01

**Decision:**

Accept (poster)

**Comment:**

I'd like to thank the authors and reviewers for great efforts writing reviews, rebuttals, and subsequent discussion, particularly with 6 reviewers assigned to this paper eventually.

Nevertheless, there's an overall consensus towards acceptance. There is one reviewer on the rejection side.

The reviewer is in fact an expert in ML privacy and their opinion should not be taken lightly. However, the reviewer has not engaged in subsequent discussion and the AC deems most of the initial concerns by the reviewer were addressed by the rebuttal. AC also deems the score of 1 (firm reject) does not match with the verbal assessment of the paper, where the weakness of the paper cannot be considered critically strong enough to overturn the positive assessment of the other 5 reviewers.

The AC recommends a solid acceptance for the paper.